# Coordination of parental performance is breeding phase-dependent in the Dovekie (*Alle alle*), a pelagic Arctic seabird

Antoine Grissot[1]*, Lauraleen Altmeyer[1,2,3], Marion Devogel[1], Emilia Zalewska[1], Clara Borrel[1,2,3], Dorota Kidawa[1], Dariusz Jakubas[1], Katarzyna Wojczulanis-Jakubas[1]

**1** Faculty of Biology, Department of Vertebrate Ecology and Zoology, University of Gdańsk, Gdańsk, Poland, **2** Université de Rennes 1, Rennes CEDEX, France, **3** L'institut Agro (AgroCampus Ouest Rennes), Rennes Cedex, France

* antoine.grissot@gmail.com

**Data Availability Statement:** The data underlying this study and the script required for result reproduction are available online at: https://doi.org/10.57745/DH9MKM.

## Abstract

Currently, parental care is becoming increasingly perceived as male and female cooperation, instead of being primarily shaped by sexual conflict. Most studies examining cooperating performance consider coordination of parental activities, and doing so focuses on a short time-window including only one stage of breeding (i.e., incubation or chick rearing period). Here, we considered the cooperation of breeding partners, investigating the coordination of parental care in a long-lived seabird species with long and extensive biparental care, the Dovekie (or Little Auk), *Alle alle*, and looked at the issue throughout the breeding season. Previous studies on this species revealed coordinated chick provisioning, but parental coordination during incubation remains unstudied. Using video recordings collected over the course of two breeding seasons, we tested whether coordination was subject to small-scale changes within each stage and whether there was a relationship between coordination levels across the two stages. We found that the level of parental coordination is overall high and increases during the incubation period but decreases through the chick rearing phases. There were some inter-annual differences in the coordination level both at the incubation and chick rearing stages. We also found some dependency between the coordination during the incubation and chick rearing periods. All these results suggest that coordination is not a fixed behavior but breeding-phase dependent. The present study thus provides insights into how parental care and parents' cooperation is shaped by brood needs and conditions. It also highlights a relationship between coordination levels during chick rearing and incubation periods, suggesting some extent of temporal dependence in coordination of parental performance within the breeding season.

## Introduction

Parental care is widely spread among animals, allowing a parent to increase its inclusive fitness through successful reproduction [1, 2]. However, investing in care is costly in terms of time

**Funding:** The study was supported by Poland through National Science Centre (no: 2017/25/B/NZ8/01417 to KWJ, and no: 2017/26/D/NZ8/00005 to DK). The funders had no role in study design, data collection and analysis, decision to publish, or preparation of the manuscript.

**Competing interests:** The authors have declared that no competing interests exist.

and energy devoted to the offspring [3] and a parent engaging into the care of the current offspring could jeopardize its survival and/or future reproduction [4, 5]. Therefore parenting individuals have to balance resource allocation between current offspring care and self-maintenance [6]. When two parents are involved in care, as is the case in 81% of avian species [7], the complexity of interactions increases and sexual conflict over the care arises (summarised by Parker *et al.* [8]). This sexual conflict is illustrated by one individual free-riding on the other's effort, as both partners benefit from joint investment in care provided to the offspring but each pays a cost of providing that care on their own [4, 9–12]. This interplay of conflicted partners may lead to lower fitness of the offspring, as demonstrated both theoretically [13] and empirically [14]. However, a cooperation manifested by coordination in parental activities was proposed as a mitigation of this conflict, with a focus on bird species with biparental care [15, 16] and reviewed by Griffith [17].

An increasing number of studies suggests that the coordination of parental performance in birds with biparental care system, such as synchronization or alternation of nest visits during the incubation or chick rearing period, may increase the breeding success [18–22]. For example, Eurasian Blackcap, *Sylvia atricapilla*, parents synchronize their feeding visits in respect to each other. This way, they minimize the disturbance around the nest and, in turn, reduce the risk of nest predation [23]. Parents adjusting their performance in respect to each other may also mutually improve their own body condition, which is essential for future reproduction [24–26]. The latter issue is particularly relevant in avian species with a long-term pair bond [20], where both partners benefit from their higher survival owing to the familiarity effect (i.e., reduced time investment in pair bonding in successive breeding seasons and/or more efficient parental care with a known partner) and partner value (i.e., long-term advantages of staying with the same partner; see Griffith [17]). Indeed, there is evidence that partners that meet earlier and/or stay together for a longer period of time exhibit better breeding performance than newly formed pairs [27, 28].

Studies on parental coordination in birds initially focused on the feeding of offspring and most of the work has been performed on passerines [18, 19, 29]. Recently, other species and stages of breeding (incubation and chick rearing) have been investigated, revealing a very broad range of species-specific behavioral patterns and mechanisms [24]. However, many studies considering parental coordination still focus on a short time-window, within a single breeding stage, rather than considering it over a longer period of time, such as a whole breeding stage or the whole breeding season. Even when a breeding stage is considered in its entirety [24, 30], changes in parental coordination over the course of the period are often overlooked. Since each breeding stage has its own specific characteristics, for instance in parental behavior [31, 32], metabolic rate [33, 34], levels of various hormones [31, 35] and body conditions parameters [36], one could expect that these characteristics also influence the coordination of parental performance.

Parental coordination at particular stages of breeding (i.e., incubation or chick rearing) may be expected to be an outcome of environmental pressures, as the parental care system evolved in response to environmental conditions [37–39]. Harsh and/or unpredictable environmental conditions impose obligatory biparental care, whereas habitats with temporarily abundant food resources make biparental care facultative and condition-dependent [7, 40–42]. Similarly, changes in environmental conditions over the course of breeding may force parents to coordinate during particular stages of breeding at a level dependent to environmental challenges. As a consequence, one may expect coordination to be higher during the incubation than the chick rearing period in some species. For example, in those living in a harsh environment, where low ambient temperature would be a key environmental variable, parents may exhibit higher coordination during the incubation than during the chick rearing period,

as a developing embryo may be more sensitive to parental neglect than a hatched chick. However, in most bird species parents will not coordinate their activities at all during the incubation period, as each of them has their sex-specific tasks, with females often incubating on their own and males guarding the nest site/territory [7]. Then, during the chick rearing, when tasks are similar for male and female parents [7], factors such as thermal dependency of the chicks, food availability and/or predation pressure may elicit parental coordination. For example, in the context of high predation pressure in the environment, parents may coordinate food delivery to the offspring in a way that minimizes disturbance around the nest, and so the risk of predation [23]. Finally, similar level of parental coordination may be expected during the incubation and chick rearing period in species where full engagement into parental care by both the female and male is crucial to raise the brood successfully [7], although drivers of the coordination may be different for each phase.

Within a breeding stage parental coordination may also change, as both needs of the brood and the environmental context change over time. During the incubation, parental coordination may be most important at a later phase of the period, as well developed embryos may be more sensitive to parent neglect than an embryo that just started to develop [43]. If this is the case, parental coordination is expected to increase over the incubation period. During chick rearing, young chicks may be thermally dependent on the parents (require brooding) as well as more sensitive to both predation and starvation than older ones. Thus, parents may better coordinate their provisioning at the early phase of the chick rearing. Indeed, some studies demonstrate that the level of parental coordination is higher at the beginning of the chick rearing than later in the season [23], but see Baldan & Griggo [44]. Then, environmental conditions related to ambient temperature and food abundance may also change during the breeding period, affecting birds' overall performance, including parental coordination. For example, piscivorous seabirds breeding in large colonies face depleted food resources over the breeding season in close vicinity of the colony (so called "Storer-Ashmole" or "halo" effect [45, 46]), which results in increased duration of the foraging trips [47–49]. Changes in duration of the foraging trips are then likely to impose changes in parental coordination of food provisioning.

For species that exhibit coordinated parental care during both incubation and chick rearing periods, an unanswered question is whether the parents are somehow consistent in their performance over the whole breeding season (but see McCully *et al.* [50]). Due to different behavioral and coordinational patterns (e.g., alternation of incubation and synchronization of nest visits to reduce predation pressure during chick rearing period), as well as levels of parental investment and associated costs during the two periods, one could expect them to be unrelated to each other. On the other hand, at the beginning of the season, parents may need time to adjust to each other and synchronise their actions. Then, physiological changes over the season may further affect the parental performance [51], leading to higher parental coordination later in the season than at its onset. Understanding this interconnection between breeding stages is an important step to fully understand mechanisms behind all the observed patterns of parental coordination.

Pelagic seabirds are top marine predators, with life-history traits that make them ideal model species for investigating cooperation in parental care. Most species are long-lived, socially and genetically monogamous with a long-term pair bond, and long, extensive biparental care. Contribution of breeding partners is usually similar in all the parental tasks (i.e., incubation, brooding, and provisioning the offspring [52]), and partner value is usually high [17]. In such a system, parental cooperation is likely to be favoured by selection [17]. Besides, as seabirds forage on food resources that are unpredictable in space and time [53], raising their offspring may be particularly costly. Therefore, both parents' contribution is essential for

successful breeding [52]. Thus, coordination of parental activities in seabirds is thought to be a key adaptation for coping with the harsh and variable environment they breed in. Indeed, it has already been demonstrated that seabird parents coordinate their parental performance [22, 24–26].

In this study, we focus on a small Arctic seabird, the Dovekie (or Little Auk), *Alle alle*, to study changes in parental coordination across the whole breeding season, and a potential interconnection between levels of coordination during the incubation and chick rearing periods. The Dovekie exhibits a life-history strategy typical to seabirds (i.e., long-lived, socially monogamous with long-term pair bond, and long and extensive biparental care over a brood of small size, here restricted to a single chick), which makes it a good model species to study coordination (reviewed in Wojczulanis-Jakubas *et al.* [54]). The coordination of parental duties in this species has been demonstrated, although solely focusing on chick provisioning during the mid phase of the chick rearing period [25, 26]. Thus here, we aim to extend previous work on Dovekie parental coordination, by investigating it both during the incubation and chick rearing periods, and analyzing its level (and consistency) both within and between the two periods.

Egg protection and thermoregulation may be crucial for hatching success in the conditions of the cold Arctic summer [55]. However, the egg has apparently some tolerance to temporary neglection (no precise data available but our personal observations suggest a few minutes to up to a couple of hours) We thus hypothesized that Dovekie parents coordinated their activities during the incubation period in a way that would maximize the presence of one parent in the nest (thus possibly also the hatching success). If that is the case, we expected that when one parent was incubating the egg (offspring maintenance), the other was foraging (self-maintenance) and they would exchange; the egg would then be continuously incubated and both partners could replenish their body reserves after a long incubation bout. We further expected a positive progression of this coordination over the course of the incubation period due to, for instance, hormonal changes [56]. Since the pelagic life-style of the Dovekie imposes long lasting foraging trips, we also hypothesized that parents would coordinate these trips during the chick rearing period in a way that optimized the food delivery rate to the chick. We further hypothesized that this coordination would be particularly important at early phase of the chick's life, when chick is not yet thermoregulatory independent (first week of life) and may be more prone to starvation [57–59]. Consistently, we expected that parental coordination in chick provisioning will be higher at early than mid chick rearing phase. Finally, expecting some sort of dependency in parental behavior between the incubation and chick rearing periods, we hypothesized that coordination established during the incubation would be related to the coordination at the later phases of breeding, with pairs highly coordinated during incubation sustaining high level of coordination also during the chick rearing period.

## Methods

### Study site and video recordings

We performed the fieldwork in two consecutive breeding seasons, 2019 and 2020 (June to August) in the Dovekie colony at Ariekammen slope (77°00′ N, 15°33′ E) in Hornsund, SW Spitsbergen. This very well-studied colony is located in one of the densest breeding concentrations of Dovekies in Svalbard (ca 590,000 breeding pairs [60]). All fieldwork was performed under supervision of KWJ and DJ (having the relevant qualifications and experience). While marking birds, we handled them for no more than 5 minutes and put them back in the nest unharmed just after the procedure. We recorded and handled the birds under permission of the Norwegian Animal Research Committee and the Governor of Svalbard (2007/00150-9,

2007/00150-11, 17/00663-2, 17/00663-7). We monitored 23 breeding pairs in 2019, and 20 out of theses 23 pairs were monitored again in 2020. We established phenology (egg laying, hatching, and fledging date) by recording nest content (i.e., egg/chick) every day for a week before an expected event, according to known phenology from previous seasons and usual length of incubation and chick rearing periods (28 and 21 days respectively for hatching and fledging [55]). In 2020, egg laying events could not be monitored, thus for both seasons hatching date was used as a reference point of bird phenology. We established breeding success based on whether or not the breeding attempt led to a successful fledging, and only included successful pairs in further analyses, resulting in 18 pairs in 2019, and 16 in 2020 (with 13 pairs monitored and successful in both studied seasons), accounting in total for 21 different pairs over the two investigated seasons. Although considering failed breeders could be helpful in understanding of coordination mechanisms, it would change the study question and our sample size would not be sufficient. Thus, with the main aim to analyze the parents' performance across the whole breeding season, we focused on successful breeders only. Hence, our inference on parental coordination dynamics applies only to successful breeding pairs.

Each parent in each pair was metal-ringed and molecularly sexed in previous seasons, and additionally marked in the study seasons by a unique combination of colour rings and a colour mark on the breast's feathers (waterproof markers, Sharpie USA). The area surrounding the monitored pair's nest was video-recorded using a separate video camera (commercial HD model of JVC, Japan) placed in front of its entrance. Chosen settings allowed to record presence/absence and behavior of focal parents within a 3 m radius from their nest entrance. This is the principal area where the birds spent their time when in the colony (based on own direct observations of bird behavior in the study colony). All recordings were made in a time-lapse mode (1 frame per sec), which allowed to capture all the birds presence and behaviors of interest, while economizing memory space and later processing of the video material.

Each pair was video-recorded for several continuous 48-hour sessions, aimed to be distributed equally over the whole breeding period. Throughout the incubation, we aimed to record three sessions per pair representing the early, mid, and late phases of incubation. However, due to the aforementioned lack of egg laying data for 2020, the recordings were performed in a slightly different way for the two years: in accordance with egg laying date of each pair in 2019 (i.e., adjusted to the pair phenology) and on fixed calendar days for all pairs in 2020 (i.e., *a priori* unknown phenology) with the incubation phase being back-calculated using the hatching date. Dovekies are highly synchronised in breeding phenology at the population level [61], and hatching dates are usually distributed over a week (Jakubas *et al.* [62] and personal observations). As a consequence, in 2020 the first session (supposed to be representing the early incubation phase) started when pairs were on average in the 20th day before hatching (min-max: 16–27 d), when it was the 26th day before hatching (min-max: 24–27 d) in 2019. The second session (mid incubation phase), on average, started on the 12th day before hatching (min-max: 8–19 d) in 2020, whereas it was the 15th day before hatching (min-max: 13–17 d) in 2019. The third session (late incubation phase) in 2020 started on the 4th day before hatching (min-max: 2–11 d), against the 5th day before hatching (min-max: 4–7 d) in 2019 (see S1 Table). Given the wider range of incubation phases present in each type of recording session in 2020, it was not fully comparable with 2019. Therefore, further analyses including both years used the relative days before hatching date as a measure of the incubation phase. During the chick rearing period in both studied years, we performed two recording sessions per pair, perfectly timed to hatching phenology, representing the early (session started when chick was 4 days after hatching on average; min-max: 3–5 d) and mid chick rearing phases (11 days after hatching; min-max: 11–12 d). The phase of the period is therefore represented by a different metric for incubation (number of day before hatching: continuous) and chick rearing period (early or mid:

categorical), but still conveys the same concept of when the recording session was performed in respect to the breeding period (early, mid, late).

The video material was then processed using VLC software (VideoLAN, France) or Quick-Time player (Apple Inc. USA). While watching the videos, we noted the time (with 1 sec precision) when focal individuals were appearing/disappearing on the frame and when they were entering/exiting the nest. We also noted the presence/absence of food in adult bird gular pouch. Videos had to be processed manually, which is time-consuming and so, for efficiency, they were split between four observers (AG, LA, CB, MD). An extensive video processing training was performed using inter-observer comparisons. For that, data from videos already processed by the lead observer (AG) were used to be compared with the output from the same videos processed by observers in training. We kept training until we reached 100% similarity of the output for few videos. The birds presence/absence and food in the gular pouch are all very conspicuous, and thus inter-observer difference in the outcome of the video watching was negligible, being solely the question of 1–2 seconds difference in the time-stamp for an event (i.e., none of the birds presence/absence with/without food was recorded differently by independent observers). Due to camera failure and/or bad quality of the framing around the nest entrance, some sessions had to be discarded, and so sample size varied slightly among the analyses (provided in details in Table 1). We extracted from the video recordings the following behavioral categories expressed by time-intervals between important events: (1) "nest"–the time interval between when a focal individual entered and exited the nest (both incubation and chick rearing period); (2) "colony"–the time interval between when a focal individual was visible in the nest vicinity but not in the nest (i.e., seen repetitively in the frame, with < 1 h gap in between each at-frame presence, both periods). Individuals, when present in the colony, spend most of their time in the surroundings of their nest (Pers. Observations). Furthermore when they leave the frame and come back with a full gular pouch, we know a foraging trip was performed. There is a possibility that they spend a little time in the colony without being seen on screen before actually departing for the foraging trip, however this time is negligible; and (3) "foraging"–the time interval when a focal individual disappeared for $\geq$ 1 h (the incubation period), or the time interval between when a focal individual left the frame and came back with a full gular pouch (chick rearing period only). We choose the threshold of 1 h for the foraging trip based on previous studies on foraging durations, where average duration of the short trips was 2.03–2.41 hours [53, 63–66]. We have also never observed in the present study a bird coming back to the colony with a food load (indicating foraging) after an absence shorter than one hour.

All data manipulations and statistical analyses were performed in R environment version 4.1.2 [67], using custom made functions or existing packages, specified in the relevant context. Statistical significance was considered at alpha level of < 0.05.

## Coordination of parental care during the incubation period

To investigate parental coordination during the incubation period, we focused on the three behavioral categories described above: "nest", "colony" and "foraging". They represent, quite accurately, main Dovekie parental activities during the incubation period. Indeed, to ensure successful development of the embryo resulting in hatching, the egg needs to be kept within viable temperatures, and therefore cannot be left unattended for too long [55]. Thus, each parent is faced with a trade-off between the need to incubate the egg (represented here by "nest" category, i.e., the time they spend in the nest) and their own need to maintain body reserves ("foraging" category, i.e., the time they spend in foraging). When considering the Dovekie pair, potential conflict could rise from both partners addressing the trade-off independently

**Table 1. Model structures and summaries for the models of coordination changes within and between breeding stages.**

| Model | Type of Model | Family (link) | Model structure | Explanatory variable | Df | Estimate | SE | stat | stat_value | p_value | N |
|---|---|---|---|---|---|---|---|---|---|---|---|
| **incubation** | *GLMM* | *Gamma (inverse)* | incubation coordination ~ incubation phase (continuous; number of days before hatching) * year + (1\|nest) | (Intercept) | 1 | 0.00000561 | 0.00000018 | Chisq | 1011.04 | **0** | 92 recording sessions from 19 nests |
| | | | | incubation phase | 1 | -0.00000005 | 0.00000001 | Chisq | 19.25 | **0** | |
| | | | | year | 1 | 0.00000032 | 0.00000027 | Chisq | 1.38 | 0.24 | |
| | | | | incubation phase: year | 1 | -0.00000005 | 0.00000002 | Chisq | 6.33 | **0.012** | |
| **chick rearing** | *LMM* | Gaussian | chick rearing coordination ~ chick rearing phase (categorical; early or mid) * year + (1\|nest) | (Intercept) | 1 | 0.48840530 | 0.10623574 | Chisq | 21.14 | **0** | 66 recording sessions from 21 nests |
| | | | | chick rearing phase | 1 | -0.45370521 | 0.15024002 | Chisq | 9.12 | **0.003** | |
| | | | | year | 1 | -0.56859031 | 0.15757306 | Chisq | 13.02 | **0** | |
| | | | | chick rearing phase: year | 1 | 0.66636030 | 0.22284196 | Chisq | 8.94 | **0.003** | |
| **early chick rearing** | *LM* | Gaussian | early chick rearing coordination index ~ early incubation coordination + mid incubation coordination + late incubation coordination | (Intercept) | 1 | 5.56551650 | 2.74963966 | F. value | 4.1 | 0.068 | 15 recording sessions from 15 nests |
| | | | | early incubation coordination | 1 | -0.00001279 | 0.00000814 | F. value | 2.47 | 0.144 | |
| | | | | mid incubation coordination | 1 | -0.00000470 | 0.00000720 | F. value | 0.43 | 0.527 | |
| | | | | late incubation coordination | 1 | -0.00001446 | 0.00001428 | F. value | 1.03 | 0.333 | |
| **mid chick rearing** | *LM* | Gaussian | mid chick rearing coordination index ~ early incubation coordination + mid incubation coordination + late incubation coordination | (Intercept) | 1 | 2.45268550 | 2.62706064 | F. value | 0.87 | 0.371 | 15 recording sessions from 15 nests |
| | | | | early incubation coordination | 1 | -0.00001996 | 0.00000778 | F. value | 6.59 | **0.026** | |
| | | | | mid incubation coordination | 1 | 0.00002316 | 0.00000688 | F. value | 11.33 | **0.006** | |
| | | | | late incubation coordination | 1 | -0.00002005 | 0.00001364 | F. value | 2.16 | 0.17 | |

Significant explanatory variables are indicated by a bold p value. Df: degrees of freedom; Estimate: unstandardised effect size indicating the relationship between the response variable and each explanatory variable; SE: Standard Error; N: sample size of specific models.

from each other (e.g., foraging at the same time and leaving the egg unattended, or simultaneously present at the colony and risking depletion of their body reserves). This conflict could be mitigated by coordination, i.e., partners doing the opposite activity in respect to each other ("nest" vs "foraging" or *vice versa*). To establish whether sexual conflict is apparent or is mitigated by coordination during the incubation, we calculated the amount of time when one partner was in the nest, while the other partner was foraging, and tested its significance by comparison with what could be expected by chance. We used a Monte Carlo randomization approach, following Wojczulanis-Jakubas *et al*. [25] and Grissot *et al*. [26], where this procedure was used for the investigation of coordinated chick provisioning. During the randomization procedure, performed separately for every recording session, we shuffled 10,000 times the observed continuous pattern of the three activity categories ("nest", "colony" and "foraging") for both male and female of each pair in each incubation session, with specific constraints (e.g., a "colony" activity always present before and after each "nest" and "foraging" activities), and then we compared the obtained randomised patterns with the originally observed pattern. In total, in the two investigated seasons we performed the procedure for 92 recording sessions of 19 different pairs. We calculated a p-value for each session, as the proportion of

randomisation iterations where the observed value (i.e., the observed proportion of time spent performing opposite activities for a given pair) was smaller than the expected random value (i.e., the proportion of time spent performing opposite activities calculated for each randomisation iteration). We assumed that if parents spend more time doing opposite activities than what is expected by chance, then they are actively coordinating their parental activities.

## Changes in parental coordination throughout the incubation period

For further analyses, the amount of time when the two parents performed opposite activities (i.e., one partner was in the nest, while the other was foraging) was used as a proxy for their coordination. We chose to use the duration *per se* instead of using the calculated p-value, or calculating an index based on our randomisation procedure (as for the coordinated provisioning, here and in Wojczulanis-Jakubas *et al.* [25], Grissot *et al.* [26]), to account for the fact that during the incubation period, the time prevalence of two main activities (i.e., incubation and foraging bouts) leaves little space for chance in the randomization procedure in the finite unit of 48 h that we considered. If coordination index during the incubation was to be calculated following the previously used procedures [25, 26], the described constraints could lead to a flattened value of the index. Instead, a crude duration of time intervals with overlapping partners activities exhibits considerable variation over the time. Importantly, the amount of time when the two parents performed opposite activities represents the coordination of parental performance well, and this is despite an increase in the duration of incubation bouts and/or foraging trips over time. This is because in the finite unit of time each parent can perform three behavioral activities ("nest", "colony" and "foraging"), with only two of them representing the needs of offspring ("nest") and themselves ("foraging"), the more they perform opposite activities satisfying one or the other need, in regard to each other, the more they are coordinating in the sense we consider in the present study.

To explore the changes in coordination of parental activities during the incubation period, we constructed a generalised linear mixed model with the amount of time partners performed opposite activities during a recording session as the response variable, and the phase of incubation (represented by the number of days before hatching, taken as a continuous variable) as well as the year and their interaction as explanatory variables. We included the year and its interaction with the incubation phase, as inter-annual meteorological and oceanographic variations between the two years [68] could lead to different incubation constraints. Given the fact that the response variable is an amount of time and thus follows a gamma distribution, we used this type of distribution, with an inverse link function within the *glmer()* function from the *lme4* package [69]. To account for pseudoreplication associated with multiple representation of the same pairs in data set, we also included pair identity as a random effect. We tested significance of explanatory variables with the *Anova()* function, using type III Wald Chi-square tests from the package *car* [70]. Assumptions of homoscedasticity and normal distribution of residuals were verified in the model by visual inspection of diagnostic plots. All details of model structure and results of this model, including p-values for each variable tested, as well as unstandardised effect sizes in the form of estimates of the models and standard errors are provided in Table 1.

## Parental coordination throughout the chick rearing period

To investigate parental coordination during the chick rearing period, we focused on chick provisioning as the main parental activity. To do so, and following Grissot *et al.* [26], we considered "foraging" and "at colony" activities, with the latter being "nest" and "colony" activities combined, as during the chick rearing period, the time spent in the nest is mainly dedicated to

chick provisioning and therefore consists of quick visits. Besides, in contrast to incubation, Dovekies in the study population exhibit bimodal foraging trip strategy during the chick rearing period, where a parent alternates between long trips (primarily to maintain their own body condition, as birds gain body mass during these trips) with a consecutive series of short trips (for chick provisioning, with some body mass loss for themselves [71]. Thus, the two types of foraging trips well represent parental trade-offs over care and self-maintenance during the chick rearing period. It has been shown previously during mid chick rearing that potential conflict generated by this situation can be avoided through coordination by the two parents, effectively avoiding the performance of long trips at the same time [25, 26]. Here, we adopted a procedure previously utilized in similar context [25, 26], and split the "foraging" activity into "short trips" and "long trips", according to the method proposed by Welcker *et al.* [63]. This method consisted in finding a threshold value of trip duration, that split data into two groups (short and long trips) of minimum sum of variances, given their log-transformed data distribution. The data were log-transformed to better separate the two groups (otherwise the division of the two groups is fuzzy and finding a cut-off point is not straightforward). This procedure was done separately for each chick rearing phase (i.e., early and mid) in each season, and the two optimal groups were obtained with a threshold at 6.1, 6.0 hours for the early chick rearing phase of 2019 and 2020, respectively, and at 8.95 and 8.55 for the mid chick rearing phase of 2019 and 2020, respectively (detailed information about mean duration of both trip categories for each chick rearing phase in the two investigated seasons is provided in S2 Table). Then, we calculated the observed within pair amount of time that one individual was performing a short foraging trip, and its partner a long foraging trip. We then shuffled with constraints ("at colony" always before and after a foraging trip) 10,000 times the chick provisioning patterns of the two partners. We calculated, for each iteration, the amount of time when one individual was performing a short foraging trip, while its partner was performing a long foraging trip within the obtained shuffled pattern and compared it with the originally observed one. We performed the procedure for 66 chick rearing recordings sessions from 21 different nests and we calculated a p-value for each session as the proportion of randomisation iterations where the observed value was smaller than the expected random value.

We then calculated the coordination index coined by Wojczulanis-Jakubas *et al.* [25] using the formula: $[obs—exp]$ x $exp^{-1}$, where *obs* is the observed amount of time with one partner performing a short trip and the other on a long trip and *exp* is the mean of all the values obtained during the randomisation procedure. For the chick rearing period, we decided to use the coordination index instead of crude values of short-long trip overlaps between the partners to reduce some "noise" in the data related to how short and long trips were calculated. To split the trips into the two types we used the method proposed by Welcker *et al.* [63], where a cut-off is applied for the continuous variable, resulting in trips of duration around the cut-off point being somehow uncertain in respect to their purpose.

We explored the changes of the coordination index over the course of the chick rearing period by fitting a linear mixed model using the *lmer()* function of the package *lme4*, with the coordination index as the response variable, and the phase of the chick rearing period (representing whether the recording session was performed in the early or mid phase of the chick rearing period, a categorical variable) as well as the year and their interaction as explanatory variables (for the same reasons as for the incubation period detailed above). To account for pseudoreplication (multiple representation of given pairs), we also included pair identity as a random effect. Significance of explanatory variables was established as for the models in the incubation period. Whenever we found qualitative explanatory variables or their interactions to be significant, we performed post-hoc Tukey tests to assess specific differences, using the *emmeans()* function from the *emmeans* package [72]. We performed the Tukey tests with all

possible pair-wise combination, despite it being conservative, to prioritise control of 1$^{st}$ type error, but we reported in the results and on Fig 2 only the biologically meaningful comparisons. Assumptions of homoscedasticity and normal distribution of residuals were also verified in the model by visual inspection of diagnostic plots. We provided details of the structure and results of this model in Table 1.

### Link between the coordination in incubation and chick rearing

We investigated a potential relationship between the coordination of the two breeding stages (i.e., incubation and chick rearing). To account for differences within the incubation period, we selected only data from 2019, for which we performed three recording sessions clearly separated in terms of phenology, therefore the phase of incubation is here represented as a categorical variable with three levels (early, mid, and late). We also included in this analysis only the pairs for which we had three incubation recording sessions successfully performed, as well as two full chick rearing recording sessions, resulting in a sample size of 15 pairs. We fitted a linear model with coordination index in the chick rearing period as a response variable, and coordination during each phase of the incubation period (i.e., amount of time partners are performing opposite activities) as explanatory variables. In order to account for differences between phases of the chick rearing period (i.e., early and mid), we fitted two separate models, one for each phase of the chick rearing period, and we included in the models parental coordination in each phase of the incubation (early, mid, and late) as independent explanatory variables. We checked the non-multicollinearity of the explanatory variables in both models by calculating the variance inflation factor (1.08, 1.07 and 1.01 for early, mid and late incubation coordination, respectively) using the *vif()* function from the *car* package. Like previously, we tested their significance with type III Wald Chi-square. We also verified assumptions of homoscedasticity and normal distribution of residuals in these models by visual inspection of diagnostic plots (S1, S2, S4 and S5 Figs in S1 File). Due to relatively small sample size and potentially influential points (Fig 3), we additionally evaluated the estimates of the models with a bootstrap procedure (results presented in S1 File). We provide structure and results of the models in Table 1.

## Results

### Parental coordination throughout incubation

During the incubation period, the amount of time when one partner was incubating in the nest while the other was foraging at sea was overall high and, on average, represented 88% (interquartile range: 79–98%) of the time of a given recording session. This value was greater than expected by chance in 91% (84 out of 92) of the investigated recording sessions, indicating that partners often spent more time than expected by chance performing opposite activities in the studied population.

The amount of time when partners performed opposite activities (i.e., a proxy of the incubation coordination) was significantly affected by the incubation phase (GLMM, gamma family, $\chi^2$ = 19.25, P < 0.001, Table 1) and its interaction with the year (GLMM, gamma family, $\chi^2$ = 6.33, P = 0.012, Table 1). The amount of time when partners performed opposite activities increased with time, being the highest just before the hatching date (Fig 1). This increase was more accentuated in 2020, compared to 2019 (Fig 1).

### Parental coordination throughout chick rearing

During the chick rearing period, the amount of time one partner was performing a long trip while the other was performing a short trip was relatively high and on average represented

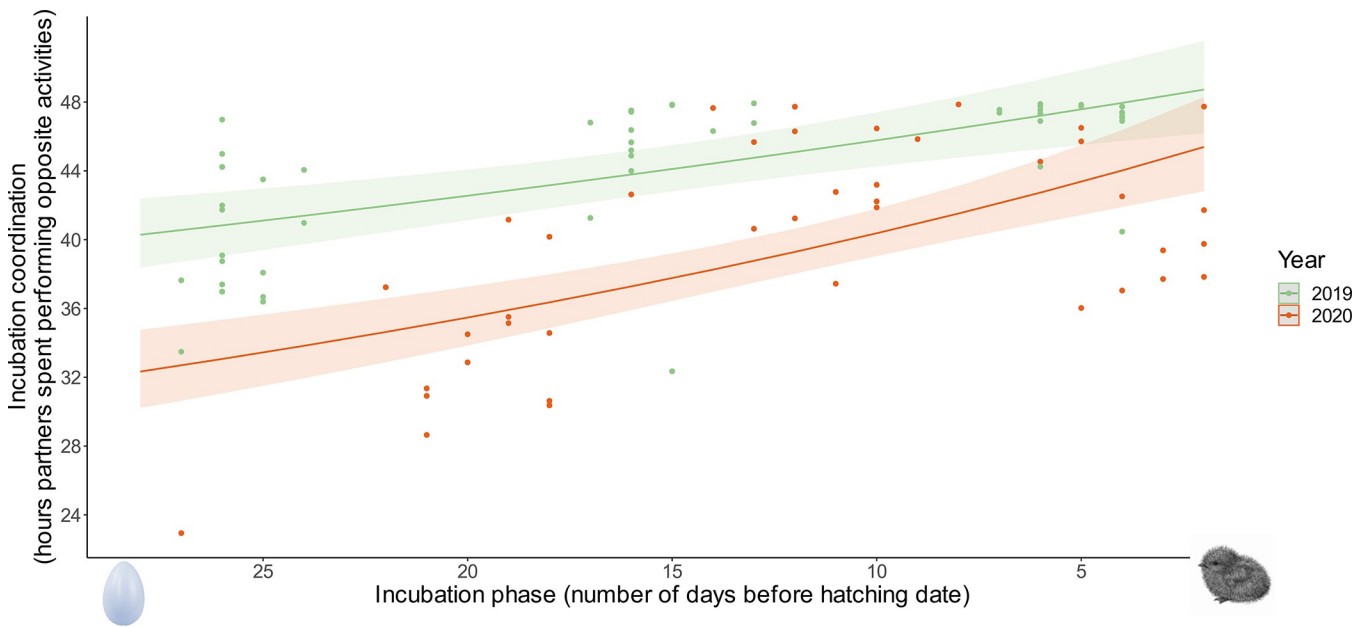

**Fig 1. Changes in the coordination of parental performance between partners during the incubation period.** The dots (with colours representing years of the study) represent a given 48 h recording session for one focal pair. The incubation coordination is the amount of time from this particular recording session that partners spent performing opposite activities (i.e., one partner incubating in the nest, while the other was foraging at sea). Incubation phase is expressed as the number of days before the hatching date of the given pair on which the recording session was performed (i.e., the lower the number, the closer to the hatching date, illustrated by an egg to the left side and a chick to the left). Lines (with particular years indicated by colours) represent the regression obtained from the candidate GLMM, with shaded areas representing 95% confidence intervals.

27% (interquartile range: 16–36%) of the time of a given recording session. Such an amount of time was greater than expected by chance in 15% (10 out of 66) of the recording sessions, indicating that coordination of long and short trips of both partners happens but is not that frequent in the studied population.

The coordination index was significantly affected by the chick rearing phase (LMM, $\chi^2$ = 9.12, P = 0.002, Table 1), as well as by the year (LMM, $\chi^2$ = 13.02, P < 0.001, Table 1), and their interaction (LMM, $\chi^2$ = 8.94, P = 0.003, Table 1). Early chick rearing was characterised by a higher coordination index compared to the mid phase in one season (2019) but rather similar in the other (2020; *post-hoc* Tukey test, with P < 0.05 for significant comparisons; see Fig 2).

### Link between the coordination in incubation and chick rearing

The coordination index of the early phase of the chick rearing period was negatively related to the way partners coordinated their activities during any of the incubation phases, and none of the estimates were significant in the standard-testing procedure (LM, F = 3.81, P = 0.07 for early incubation, F = 0.99, P > 0.05 and F = 1.02, P > 0.05 for the mid and late incubation phases respectively, Table 1). The estimates for early and late incubation became significant, however, when the bootstrap procedure was applied in the modelling (S3 Fig in S1 File). At the mid phase of the chick rearing period, the coordination index was significantly related to the coordination at the early (LM, F = 6.91, P = 0.02, Table 1) and mid phases of the incubation (LM, F = 10.95, P < 0.01, Table 1). The direction of the relationship was the opposite for these two phases (see Fig 3), with the pairs highly coordinated during chick rearing exhibiting a lower level of coordination during early incubation but a higher level of coordination in mid incubation (Fig 3). The estimates established with the bootstrap procedure remained of quite

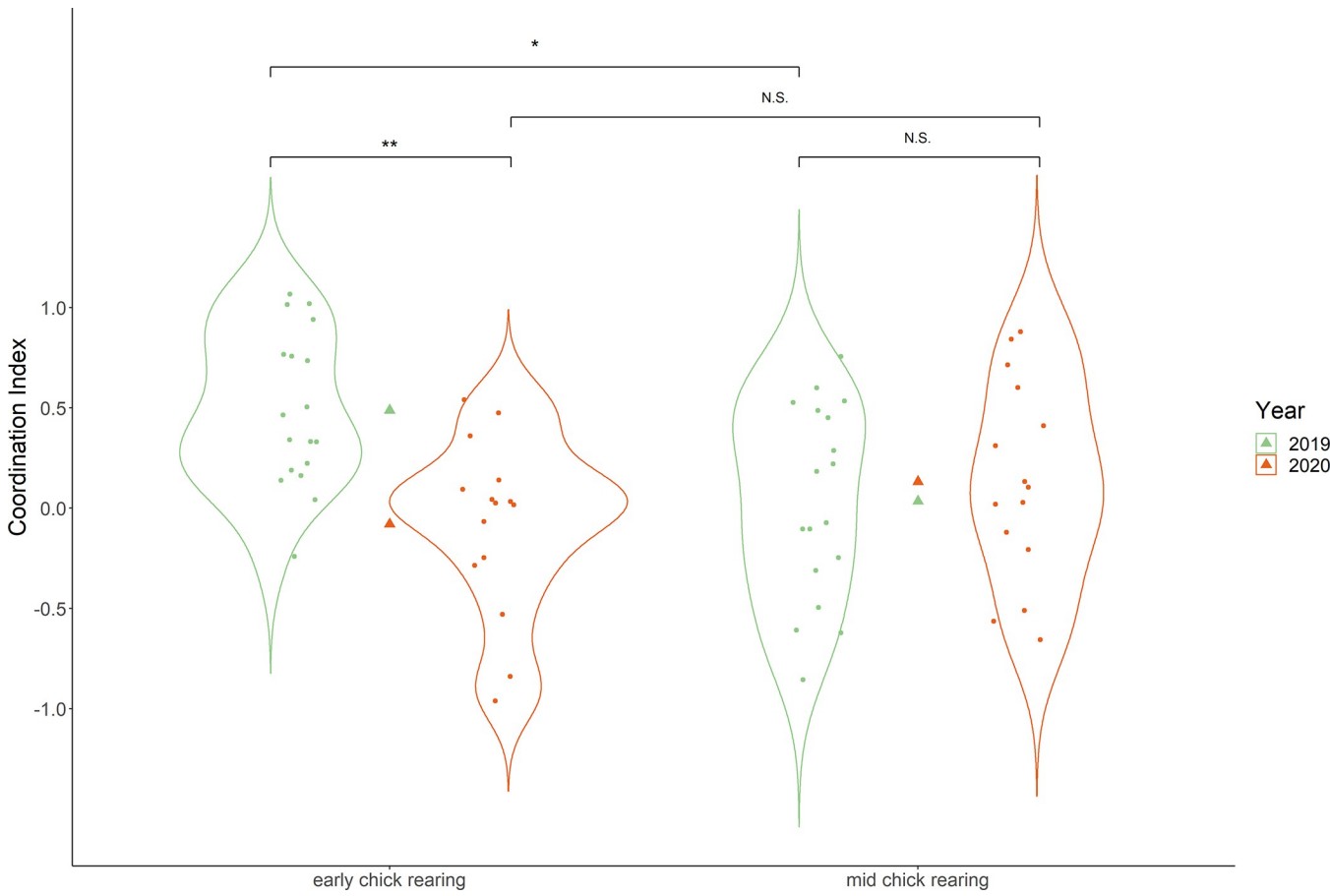

**Fig 2. Differences in the coordination index during early and mid phases of the chick rearing periods in two consecutive years.** Values of the coordination index above 0 indicate coordination, i.e., considerable overlap of short and long foraging trips of the partners. Violin plots represent the distribution, dots the coordination value for each pair, and triangles the mean for the phase in given year (indicated by color). Difference between every combination was tested with a pair-wise *post-hoc* Tukey test, and significance is indicated at the top of black horizontal lines. (N.S.: P > 0.05; *: P < 0.05; **: P < 0.01; ***: P < 0.001).

similar values and significance as in the standard approach, with the exception for mid incubation, where the value became marginally significant (P = 0.08, S6 Fig in S1 File).

## Discussion

We showed that Dovekie partners coordinate their performance in respect to each other throughout incubation and also tend to do it during the chick-rearing period, with specific patterns in each breeding stage corresponding to the different constraints imposed on the reproduction attempt. Our results provide an insight into the way that parents optimize their parental performance in a long-lived monogamous seabird. Furthermore, our results reveal changes in coordination within the course of each studied breeding stage and in response to the year, suggesting that biparental care performance, even in strictly biparental care system, might not be a fixed *sensu* ("sealed bid" as suggested by Houston & Davies [9]) but a flexible behavior, even though additional studies on the matter are needed.

Cooperation in parental care is receiving growing and well deserved attention but few studies take into consideration more than one breeding stage at a time. Here, our study species was already known for its coordination of the chick provisioning [25, 26] but by being examined in the mid chick rearing period only. Then, the incubation period remained unstudied in the

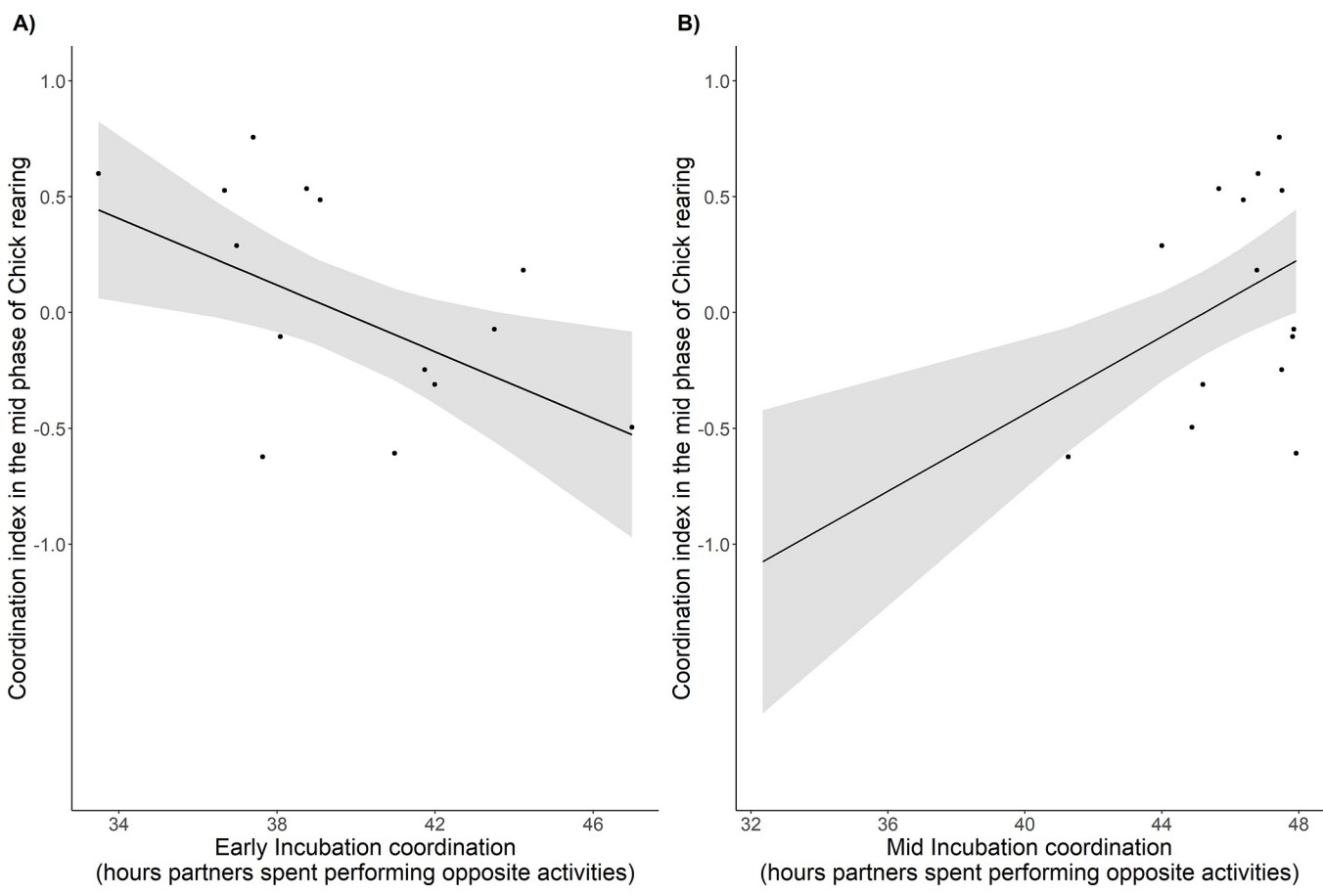

**Fig 3. Relationship between coordination during incubation and the coordination index during the mid phase of the chick rearing period.** The dots represent a focal pair. The incubation coordination is the amount of time from the early (A) or mid (B) phase of incubation that partners spent performing opposite activities (i.e., one partner incubating in the nest, while the other was foraging at sea). Lines represent the regressions obtained from the candidate LM, with shaded areas representing the 95% confidence intervals.

same context. We bridged this gap by showing that Dovekie parents spend more time than expected by chance performing opposite activities during the incubation period (i.e., one partner incubating the egg, and the other foraging), which is likely to be an effect of their active coordination of parental activity. Concerning the coordination of chick provisioning, initially highlighted by Wojczulanis-Jakubas et al. [25] and Grissot et al. [26], we showed here that some parents during the chick rearing period actively alternate their short and long trips in respect to each other but this is apparently not so straightforward, as only some parents seem to be able to coordinate better than by chance. Additionally, the level of this coordination is also season- and phase-dependent (higher in early chick rearing in some seasons). The fact that coordinated parental care is exhibited by the Dovekie during both incubation and chick rearing indicates that both stages present their own constraints and parents are apparently able to mitigate these effects through the coordination of their parental activities. Such coordination throughout the whole breeding season is not surprising, given high energy requirements in Dovekies [58, 73, 74], harsh breeding environment [55], including quite challenging and unpredictable foraging conditions [26, 53].

We showed that within incubation and chick rearing periods, Dovekie pairs exhibit changes in coordination levels. Those changes seem to be going in opposite directions in the two

examined breeding stages, as coordination during incubation is on average increasing over the course of this stage (Fig 1), and during the chick rearing period, coordination is higher in early chick rearing than the mid phase of the same period (Fig 2), even though this pattern seems to be season dependent. This is, in general, in accordance with our initial hypotheses but direct factors shaping the patterns remain unknown, and require further investigation. For the time being, we can only suggest some intrinsic drivers and mechanisms. During incubation, there could be a growing need of the developing embryo for a constantly high incubation temperature, thus increasing parental coordination could favour embryo development. For chick rearing, increasing thermal independence and decreasing sensitivity to starvation of the chick as it ages [57, 58] might decrease the need for parental coordination. Hormonal changes in adults are likely to play a role in the regulation of parental behavior, especially during the incubation period. It is, for instance, known that the level of prolactin increases over the course of the incubation period in the Dovekie parents [35, 56] but whether this increases their propensity to incubate, and therefore their level of coordination, remains unknown and will require specific studies.

Although we did not aim to examine inter-annual differences in parental behavior, conducting the study across two breeding seasons, we considered the effect of the year and its interaction with the different phases of breeding stages. Interestingly, in both breeding stages we found some interannual differences (see Table 1 and Figs 1 and 2). During the incubation period, although we found coordination to increase in the course of the period in both studied seasons, the intercept for the increase differed between the seasons, indicating that Dovekie parents were more coordinated in their incubation activities in one season (2019) compared to the other (2020). During chick rearing, we found that chick provisioning was more coordinated in early chick rearing compared to the mid phase in one season (2019) but not the other (2020). Putting all these results together, we can conclude that in the 2019 season, birds were more coordinated than in 2020. With only two seasons being examined, without properly investigated environmental context, it is hard to interpret inter-annual differences in the observed patterns. We can only speculate that environmental conditions may affect parental coordination, as was shown in other species (*Charadrius* spp.) relating to biparental care [75] where parental coordination increased with temperature stochasticity.

Grissot *et al.* [26] suggested some effects of environmental conditions on parental coordination during the chick rearing period, even though no clear inter-year differences could be highlighted in that study. Given that the two studies were conducted in the same species and colony, with similar methodologies, it is intriguing that we could highlight in the present study an inter-annual variation that was not highlighted in the previous one. Based on the suggestion made that the range of environmental conditions was not extreme enough to highlight changes in coordination strategy in the previous study, we could hypothesize that years investigated in the present study represented a wider range of environmental conditions. Another explanation for the inter-annual differences in coordination revealed in the present study could come from the greater range of temporal scale used here. Grissot *et al.* [26] focused solely on coordination during the mid phase of the chick rearing period, which seems to be the least subject to inter-year differences in the present study (Fig 2). Given that coordination of parental performance during chick rearing is decreasing between the early and mid phases, and that breeding constraints (e.g., chick thermoregulation and feeding) are at their highest level [55], one could assume that the coordination of chick provisioning is therefore more crucial at the early phase, just after the hatching date. Therefore, coordination at mid chick rearing seems to influence the breeding outcome less, which is also concordant with the apparent lack of significant effects of coordination at the mid chick rearing period on the chick growth rate, as reported by Wojczulanis-Jakubas *et al.* [25] and Grissot *et al.* [26]. All these findings clearly show that examining parental coordination in different breeding stages (and different environmental

contexts) allows to capture fine-scale changes in parental behavior and is important for recognizing all the factors influencing parental cooperation.

Our results indicated the existence of some relationships between parental coordination during the incubation and chick rearing stages (Fig 3). However, given the instability of the results between linear modelling and the bootstrap procedure (S1 File), the opposite directions of relationships, and the fact the link between the stages has been understudied in general, the results are hard to interpret. We found no effect of incubation coordination on the coordination index of the early phase of chick rearing, although the relationships (negative) became significant in an alternative bootstrap procedure. With such differences, we do not even try to interpret these findings. The relationship between the coordination index of mid chick rearing and early and mid-incubation coordination was stably significant, however, with opposite direction for the early and mid phases. To interpret this, one possibility is that parental coordination during mid chick rearing, is dominated by parental constraints rather than being related to brood needs and might reflect pair quality. Then, early and mid-incubation periods, during which partners are in the process of re-bonding and habituating to parental activities, seem to be dominated only by parental constraints rather than the chick's, and parental coordination could also reflect pair quality. The link found between mid chick rearing and early/ mid incubation substantiate this argument, and would thus inform us about partners' communication and/or pair quality. How to interpret this quality is a separate question. Coordination during late incubation, in that context, being the most important from the perspective of a developing embryo, would mask the pair-quality effect. Such reasoning about the functionality of coordination during late incubation could be supported by the very high level of the coordination reached by all the pairs, and low inter-pair variation (Fig 1). Clearly, the pair-quality-based interpretation is very speculative and further investigation on the link between coordination during incubation and chick rearing, with a bigger sample size and/or an experimental approach is apparently needed. We also cannot exclude the possibility that revealed relationships are random and do not have any biological meaning.

Our study is based on successful breeders only, and for those we found some evidence of parental coordination, both during incubation and chick rearing. However, further studies are needed, possibly with experimental approaches, to properly examine the consequences of parental coordination on reproductive success. It is possible that parental coordination during the incubation period is the most important component of reproductive success, as there is quite high inter-pair variability in hatching success (e.g., hatching success for different sub-colonies 66–95% [76]). Once the chick hatches, its chances for successful fledging are high (e.g., chick survival up to 14 days, for different sub-colonies: 85–100% [76]). Thus, examining parental coordination at the early incubation phase along with hatchability could provide valuable insight into the significance of parental coordination.

To conclude, our study highlights the importance of considering coordination not only during a short time-window of a specific breeding stage, but also using a broader temporal scale. Indeed, Dovekie adults adjust their parental behavior and coordination with their mate throughout the breeding, possibly to match the offspring's needs. However the level of coordination within each breeding stage is influenced by breeding/environmental conditions. The exact drivers of parental coordination remain to be identified, but future studies should consider the importance of temporal changes in parental behavior.

## Supporting information

**S1 File. Supplementary materials.** Containing detailed information on egg development phases and age of the chick during recording sessions, on the threshold used for short trip/

long trip classification, as well as on the bootstrap procedure used to evaluate the estimates of the models.
(DOCX)

**S1 Table. Egg development phases and age of the chick.** Values are expressed in number of days before (resp. after hatching date), at the beginning of the recording session.
(TIF)

**S2 Table. Threshold used for the classification of short and long trips during the chick rearing period.** Values are expressed in number of hours.
(TIF)

## Acknowledgments

We thank Martyna Cendrowska for her help in the field, the Polish Polar Station for hosting us even in pandemic times, and Łukasz Pracki for his precious IT help in the field. We also thank Jacob Ligorria for proof-reading the manuscript and, as a native English speaker and specialist in ecology, for the substantial improvements he made.

## Author Contributions

**Conceptualization:** Antoine Grissot, Lauraleen Altmeyer, Dariusz Jakubas, Katarzyna Wojczulanis-Jakubas.

**Data curation:** Antoine Grissot, Lauraleen Altmeyer, Marion Devogel, Emilia Zalewska, Clara Borrel, Katarzyna Wojczulanis-Jakubas.

**Formal analysis:** Antoine Grissot, Lauraleen Altmeyer, Katarzyna Wojczulanis-Jakubas.

**Funding acquisition:** Dorota Kidawa, Katarzyna Wojczulanis-Jakubas.

**Investigation:** Antoine Grissot.

**Methodology:** Antoine Grissot, Lauraleen Altmeyer, Dariusz Jakubas, Katarzyna Wojczulanis-Jakubas.

**Resources:** Dorota Kidawa, Dariusz Jakubas, Katarzyna Wojczulanis-Jakubas.

**Writing – original draft:** Antoine Grissot.

**Writing – review & editing:** Antoine Grissot, Lauraleen Altmeyer, Marion Devogel, Emilia Zalewska, Clara Borrel, Dorota Kidawa, Dariusz Jakubas, Katarzyna Wojczulanis-Jakubas.

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
