## [Decision Letter · Decision Letter 0]

15 Jan 2024

PONE-D-23-40200Coordination of parental performance is breeding phase-dependent in the Dovekie (Alle alle), a pelagic Arctic seabirdPLOS ONE

Dear Dr. Grissot,

<o:p></o:p>

Thank you for submitting your manuscript to PLOS ONE. After careful consideration, we feel that it has merit but does not fully meet PLOS ONE’s publication criteria as it currently stands. Therefore, we invite you to submit a revised version of the manuscript that addresses the points raised during the review process.

Dear Dr., Antoine Grissot

Thank you for submitting your manuscript to PLOS ONE. After careful consideration, we have decided that your manuscript needs Major Revision.

Kind regards,

Prof. Lamiaa Mostafa Radwan, Ph.D.

Academic Editor

PLOS ONE

Editor Comments:

1- The manuscript needs Editing language

2- Material and methods need more clarity

3- Discussing results requires citing references that explain those results

**Reviewer1**

It is possible to follow a logic between methods, results and discussion, The authors are careful in the design, even when the Dovekie is broadly studied this work offers different approach.

This work follows the statistical analysis and design according to the characteristics of behavior, reproductive performance, and breeding sites characteristics.

All data is available in the manuscript or supporting materials.

On the line 309 the authors mentioned about inter-annual meteorological and oceanographic variables between years to construct the generalized liner mixed model, but little is mentioned about these variables in the results section, however on the discussion the author mentioned the importance of environmental variables.

Author must review references. Line 50 Parker et al 2002; Line 496 Wojczulanis-Jakuba et al (2018) are not listed. Otherwise on line 704 Jakubas, D., Wojczulanis-Jakubas, K., & Kreft, R. (2008). Sex differences in body condition and hematological parameters in Little Auk Alle alle during the incubation period. Ornis Fennica, 85, 90–97 is on references but not on manuscript.

**Reviewer2**

This manuscript explores coordination of parental effort for Dovekies breeding in Svalbard. The study adopts a novel whole-season perspective, whereby the authors consider coordination throughout the entire breeding period and investigate correlations between incubation and chick-rearing, two distinct periods of care underlain by different parental behaviours. While the study poses an intriguing question, and acknowledging the authors' comprehensive approach, I have concerns. The writing is a little raw and difficult to follow, especially in the methods section, where complex approaches lack full explanation. Many conclusions in the discussion seem weakly supported, with unclear logical chains, and insufficient appraisal of the wider literature. However, my major concern is that the authors may not be measuring coordination during incubation accurately.

The defined metric, 'time parents spend on opposite activities,' seems flawed, as continuous egg incubation means parents must nearly always be engaged in opposite activities. As a result, this metric instead seems to just measure respective foraging time/nest attendance, not coordination. Furthermore, I question the randomization procedure, as the null hypothesis represents a pattern of behaviour that fundamentally cannot happen – i.e., two parents behave with no regard to one another, which would leave long periods of egg neglect or overlap at the nest. This is particularly problematic considering that the authors only use successful nests, which presumably are much less likely to have neglected the egg – the only behaviour that would give rise to ‘low’ coordination in this metric. This is a valuable dataset, but I recommend a revised analytical approach – for example, by comparing the nest shifts/trip durations of each parent, as has been done in many other species. Alongside this, I would like to see clearer methods, and a more cautious interpretation of the results.

As a general point, I note quite a few issues with the language usage. In parts, this made it difficult to follow the text and it took me significant effort to parse the meaning. I recognize the complexity of writing in a second language and appreciate the authors' efforts. However, improving this will significantly benefit the overall flow and comprehension and ensure the paper is understood and read by a wide audience. I would strongly suggest that a proficient English speaker is involved in the revision process to ensure that the intended meaning is effectively conveyed, particularly as PLOS ONE does not offer copy editing. I hope that the Editor can facilitate this.

Below I have outlined my thoughts in more detail. I hope this helps the authors improve their manuscript, and I look forward to seeing an updated version.

Introduction

The introduction contains the required theory to understand the background of the study, which is well justified. However, I’m afraid I feel it needs work. The presentation of theory is quite disorganised, with several vague explanations and examples. Additional citations, concrete examples, and supporting details are needed in various parts. Similarly, the study's description at the end is unclear, especially regarding the definition of coordination in this context.

One issue is that the context and focus of the introduction is not well established – it is unclear whether it focuses on all caregiving animals, species with biparental care, or just long-term monogamous parents. The entire introduction seems bird-centric, yet this isn't explicitly acknowledged. Clarifying the focus and explicitly setting the scene is necessary to improve clarity. For example, it might be easiest to explain that the introduction is focusing on birds that are long-term monogamous, and then introduce the relevant literature in a clear way.

My line-by-line comments are follows:

L42-45: This sentence sets up a false contrast. The time and energy devoted to offspring is costly precisely because it impacts survival and/or future reproduction. If there was no future survival or reproduction (i.e., the species is semelparous), the cost of parental investment wouldn’t matter. Furthermore, I don’t think ‘evolutionarily’ is the right word here – this is describing an individual trade-off. Finally, ‘somehow jeapordizes’ makes it sound like you don’t know how – I suggest removing ‘somehow’.

L50-56: The theory is oversimplified here. Cooperation itself isn’t a solution to sexual conflict because it is evolutionarily unstable. A cooperating pair is always vulnerable to one partner free-loading on the other – there needs to be enforcement for it to be maintained. For long-term monogamous species, this enforcement may come about through the intrinsic benefit of retaining the partner – for example, because there is a high cost to divorce. Cooperation emerges only when (1) parents cannot provide uniparental care and (2) there is a high cost to losing that partner. This concept of ‘partner value’ is reviewed in Griffith 2019.

L59-60: For balance, it is probably worth noting that some studies suggest/show that sexual conflict between the parents can mean that offspring end up receiving less investment in biparental care than would be expected given the sum of the parents’ possible individual contribution, e.g.:

McNamara JM, Houston AI, Barta Z, Osorno JL. 2003. Should young ever be better off with one parent than with two? Behavioral Ecology. 14(3):301–310. doi:10.1093/beheco/14.3.301.

Royle NJ, Hartley IR, Parker GA. 2002. Sexual conflict reduces offspring fitness in zebra finches. Nature. 416(6882):733–736. doi:10.1038/416733a.

L63: I’m not sure the Tyson paper shows a positive relationship between parental body condition and coordination.

L66: It’s unclear what is meant by ‘short time of pair-bonding’

L84: I suggest ‘likely to be /a/ prime driver’ (rather than /the/) – ‘the’ suggests that this is the only driver of parental behaviour, which I suspect is unlikely.

L84-87: I’m not sure what these sentences mean. Is this saying that environmental variables might affect coordination, and might additionally affect coordination differently during different breeding periods? And if this is the case, then we might have different expectations of coordination depending on environmental context?

L88-91: This needs some citations and examples.

L91-92: Why would coordination be more pronounced during chick rearing? There is an example given for incubation but not here.

L93-97: This section really needs some references and examples to support it – it’s quite vague and unspecific. I would also like to see some percentages supporting the statement that most parents don’t coordinate during incubation (I don’t disagree, but you need to substantiate this). Similarly you need to substantiate the claim that tasks are generally more similar between the parents during chick-rearing

L102: Contribution of both parents to care does not necessarily equal coordination. For example, if the female takes on the whole of incubation, and the male takes on the whole (or part) of chick-rearing, they are both contributing to/engaging in care, but no coordination is required.

L108: Is this a general phenomenon? Or are there specific species where this has been tested?

L117: Some authors have indirectly investigated this – e.g. McCully et al 2022 find similar levels of coordination in incubation and brooding:

McCully FR, Weimerskirch H, Cornell SJ, Hatchwell BJ, Cairo M, Patrick SC. 2022. Partner intrinsic characteristics influence foraging trip duration, but not coordination of care in wandering albatrosses Diomedea exulans. Ecology and Evolution. 12(12). doi:10.1002/ece3.9621.

L117-119: This is very vague – please explain fully why they would differ.

L120: What is meant by a runway?

L121: Is there actually evidence for such a short-scale familiarity effect? My understanding is that the familiarity effect moreso relates to the idea that you don’t have to learn a new individual’s behavioural each year (i.e., is more of an annual phenomenon)

L127: I would argue that most species are not genetically monogamous. Extra-pair copulation is extremely common, including in seabirds.

L129-130: It needs to be made clear that cooperation is favoured due to partner value, not similarity in contribution – this is a circular argument.

L152-153: My understanding is that in little auks, the egg is never left unattended. So does this not mean that coordination in this context is a given, i.e. birds are constrained to wait for their partner at the nest? As a general related point, it’s quite unclear what is actually being measured in the study and what the response variables are.

L159: What does ‘habituation to the parental mode’ mean?

L161-162: What is optimized food delivery rate?

L168: Missing the word ‘period’ after incubation (or else remove ‘the’)

Methods

The methods are, in many parts, very difficult to follow. I have done my best to draw a reasonable conclusion about the approaches but there are several sentences that need to be rephrased. Additionally, as discussed above, I am not sure that the methods outlined here actually measure coordination during incubation.

L180: How was an ‘expected event’ determined?

L186-187: I appreciate the intention between only focusing on successful breeders, but there is an interesting question about whether failed breeders coordinate less (which is even mentioned in the discussion), and so I wonder why there is no analysis of this –some sort of simplistic approach that ends up in the Supplementary Materials might be interesting from the perspective of future study, even if it comes with caveats that mean it has to be cautiously interpreted.

L205-208: If I’ve interpreted this correctly, does this mean: in 2019, recordings were taken at fixed intervals following egg laying, and due to the lack of lay dates in 2020, the recordings were taken at fixed calendar dates across the colony (i.e. all birds have the same recording dates)?

L210-212: It would be useful to know how long incubation and chick rearing last for in this species.

If lay date was unknown, how were the recording days determined? It says here ‘days before hatching’ was used, but surely hatching date was unknown if lay date was unknown (at the time of planning the recordings)?

L221: ‘well-adjusted to respective phenology’ – what does this mean? Perhaps ‘timed to hatching phenology’?

L226: I think ‘precision’ rather than ‘accuracy’

L228: Were all videos watched by all observers? Or were they split between observers? If they were split, did you conduct any analysis to look for observer bias?

L229-232: Are there stats to substantiate this, e.g. in the supplementary?

L242-242: Is it possible that the individual could be present at colony for a proportion of this time? Does this matter?

L256: I’m not sure what is meant by constraints. It’s probably enough to say that they represent the main parental activities during incubation.

L258-259: See my earlier comments: if the egg needs to be incubated continuously, how can coordination as established in the introduction exist?

L268-270: If the egg is never left attended, then surely there are very rarely situations when this isn't the case? This seems to be set up as the null hypothesis, but I don't think this is realistic as we know biologically this is always the case, so how can there ever be a situation where coordination is not observed? This approach makes sense for chick-rearing where the requirement for overlap is not so strict, but I'm not sure this approach makes sense for incubation, unless egg neglect is very common.

L279-280: What is the 'observed value'? I’m not sure what is actually being measured here?

L281: Can parents ever do the same activity? If they have to incubate the egg continuously, I'm not sure how one could observe coordination in this context (except maybe in chick rearing). This approach seems to be comparing parental behaviour to something that never happens.

L288: I think this means a correlation between foraging and nest time - this makes more sense and matches previous literature.

L288-293: I’m not sure I follow this.

L295-303: I also don’t follow this.

L305-306: I don't think this is measuring coordination - it sounds like this is just measuring trip duration/nest attendance, because parents always have to be conducting opposite activities

L306-307: Minor point: phase of incubation seems a bit misleading as this is a continuous numeric variable. Perhaps just day of incubation or days until hatching?

L311: What is the distribution? I’m still not clear on what the response variable actually is.

L317: Please explain that verification was ‘by visual inspection of diagnostic plots’ (my interpretation based on the supplementary).

L343: What is a ‘chick-rearing phase’?

L347-351: On one hand, I think the short and long trips for each parent are shuffled in time and compared to the original pattern - this makes sense. But I'm not sure what the 'observed within-pair amount of time...' is.

L357: Mean of what values?

L358-359: I don’t think I understand this, because there doesn’t seem to be a coordination index for incubation.

L366-367: The phase of chick rearing should be explained much earlier on. Why is it split in this way rather than for example using days since hatching (a numeric value), as is done for incubation?

L385: Why not include the phases in the model? Given separate models are fitted, what corrections are applied to account for multiple hypothesis testing?

L386-387: Earlier, phase seems to be days until hatching (numeric), but here it's categorical. In the figures it seems to be numeric. How are these categories chosen and why is this approach used?

Results

L417-419: Perhaps I’ve misunderstood this, but I think it would be more interesting to report how frequently they take trips that are opposite, as opposed to representing this as a percentage time of the recording.

L420: Is 15% a high enough value to claim that coordination is occurring? This seems very low.

L435-436: This seems to be a post-hoc test and not justified in the methods, though I may have missed something. I am a bit concerned about this alternative approach and why the authors have used it – it feels a little like searching for a significant result.

Discussion

I feel the conclusions in the discussion are not always supported by the results, and the inferential chains aren’t always logical. I can see this difficulties in interpreting quite a mixed picture of results, and I wonder if this is partially because coordination is measured during incubation in such a different way to chick-rearing. Generally speaking, the discussion needs to be much better rooted in the published literature. There is very little reference to the wider literature, and the literature cited is a bit too dependent on the authors' previous work (while of course acknowledging that much of this lays the foundation for this study).

L486-487: I don't disagree that the behaviour is flexible, but I don't think this is something that was actually analysed here.

L493-495: I don’t agree with this interpretation – but see my previous comments.

L499-500: Different level is vague and not really what is meant - coordinate better than by chance, I think.

L501-504: This is quite a strong statement when two sentences prior, the authors make the point that there is limited evidence for coordination in chick rearing.

L507: If coordination were integral to this, surely we would expect (and find) quite strong evidence for it?

L511: I don’t think ‘as’ makes sense; the second part of the sentence doesn’t follow from the first part.

L513: ‘As we expected’ – where was this prediction made?

L514-515: I don't see anywhere where it was hypothesised that coordination would depend on the season?

L523-525: This is getting the causation backwards. Hormonal changes are a mechanistic explanation for the behaviour, not an ultimate one. In other words, hormone changes have likely evolved to facilitate the coordination pattern itself, the coordination doesn't emerge because of the hormone changes.

L526: Indeed, I don’t believe you can examine inter-annual differences with just two years, especially with different methodologies.

L535-536: I'm not sure how this conclusion is reached. How is efficiency measured?

L539: Can this be related back to the literature e.g. examples where this is the case?

L543-545: I don't think there is much evidence for this argument, and I don't think it adds anything.

L546: This is still only two years, and chick rearing is short - I doubt the temporal scale makes much difference.

L550-551: But according to the authors coordination is stronger during incubation? So why is coordination not established then (rather than after hatching)?

L560-561: This seems to be the first reference to ‘instability’ – what does this mean? (Same problem on L565).

L570: Not clear what ‘the same’ refers to.

L571-572: This needs to be better substantiated. How would a link between mid-chick rearing and mid incubation tell us anything about quality? What evidence is there for this?.

**Reviewer 3**

This study is well designed, and the manuscript provides valuable insights into how parental performance in a long-lived, monogamous seabird, emphasizing the flexibility of bi-parental care behavior rather than a fixed approach. It successfully navigates through the complexities of seasonal, stage-dependent, and inter-annual variations in parental coordination, contributing to the broader understanding of bi-parental care in seabirds.

We look forward to receiving your revised manuscript.

Kind regards,

Lamiaa Mostafa Radwan, Ph.D.

Academic Editor

PLOS ONE

 [The study was supported by Poland through National Science Centre (no: 2017/25/B/NZ8/01417 to KWJ, and no: 2017/26/D/NZ8/00005 to DK).].  

5. Please provide a complete Data Availability Statement in the submission form, ensuring you include all necessary access information or a reason for why you are unable to make your data freely accessible. If your research concerns only data provided within your submission, please write "All data are in the manuscript and/or supporting information files" as your Data Availability Statement.

6. Ethics statement only appears at the end of the manuscript:

Your ethics statement should only appear in the Methods section of your manuscript. If your ethics statement is written in any section besides the Methods, please move it to the Methods section and delete it from any other section. Please ensure that your ethics statement is included in your manuscript, as the ethics statement entered into the online submission form will not be published alongside your manuscript.

Additional Editor Comments:

Dear Dr., Antoine Grissot

Thank you for submitting your manuscript to PLOS ONE. After careful consideration, we have decided that your manuscript needs Major Revision.

Kind regards,

Prof. Lamiaa Mostafa Radwan, Ph.D.

Academic Editor

PLOS ONE

Editor Comments:

1- The manuscript needs Editing language

2- Material and methods need more clarity

3- Discussing results requires citing references that explain those results

Reviewer1

It is possible to follow a logic between methods, results and discussion, The authors are careful in the design, even when the Dovekie is broadly studied this work offers different approach.

This work follows the statistical analysis and design according to the characteristics of behavior, reproductive performance, and breeding sites characteristics.

All data is available in the manuscript or supporting materials.

On the line 309 the authors mentioned about inter-annual meteorological and oceanographic variables between years to construct the generalized liner mixed model, but little is mentioned about these variables in the results section, however on the discussion the author mentioned the importance of environmental variables.

Author must review references. Line 50 Parker et al 2002; Line 496 Wojczulanis-Jakuba et al (2018) are not listed. Otherwise on line 704 Jakubas, D., Wojczulanis-Jakubas, K., & Kreft, R. (2008). Sex differences in body condition and hematological parameters in Little Auk Alle alle during the incubation period. Ornis Fennica, 85, 90–97 is on references but not on manuscript.

Reviewer2

This manuscript explores coordination of parental effort for Dovekies breeding in Svalbard. The study adopts a novel whole-season perspective, whereby the authors consider coordination throughout the entire breeding period and investigate correlations between incubation and chick-rearing, two distinct periods of care underlain by different parental behaviours. While the study poses an intriguing question, and acknowledging the authors' comprehensive approach, I have concerns. The writing is a little raw and difficult to follow, especially in the methods section, where complex approaches lack full explanation. Many conclusions in the discussion seem weakly supported, with unclear logical chains, and insufficient appraisal of the wider literature. However, my major concern is that the authors may not be measuring coordination during incubation accurately.

The defined metric, 'time parents spend on opposite activities,' seems flawed, as continuous egg incubation means parents must nearly always be engaged in opposite activities. As a result, this metric instead seems to just measure respective foraging time/nest attendance, not coordination. Furthermore, I question the randomization procedure, as the null hypothesis represents a pattern of behaviour that fundamentally cannot happen – i.e., two parents behave with no regard to one another, which would leave long periods of egg neglect or overlap at the nest. This is particularly problematic considering that the authors only use successful nests, which presumably are much less likely to have neglected the egg – the only behaviour that would give rise to ‘low’ coordination in this metric. This is a valuable dataset, but I recommend a revised analytical approach – for example, by comparing the nest shifts/trip durations of each parent, as has been done in many other species. Alongside this, I would like to see clearer methods, and a more cautious interpretation of the results.

As a general point, I note quite a few issues with the language usage. In parts, this made it difficult to follow the text and it took me significant effort to parse the meaning. I recognize the complexity of writing in a second language and appreciate the authors' efforts. However, improving this will significantly benefit the overall flow and comprehension and ensure the paper is understood and read by a wide audience. I would strongly suggest that a proficient English speaker is involved in the revision process to ensure that the intended meaning is effectively conveyed, particularly as PLOS ONE does not offer copy editing. I hope that the Editor can facilitate this.

Below I have outlined my thoughts in more detail. I hope this helps the authors improve their manuscript, and I look forward to seeing an updated version.

Introduction

The introduction contains the required theory to understand the background of the study, which is well justified. However, I’m afraid I feel it needs work. The presentation of theory is quite disorganised, with several vague explanations and examples. Additional citations, concrete examples, and supporting details are needed in various parts. Similarly, the study's description at the end is unclear, especially regarding the definition of coordination in this context.

One issue is that the context and focus of the introduction is not well established – it is unclear whether it focuses on all caregiving animals, species with biparental care, or just long-term monogamous parents. The entire introduction seems bird-centric, yet this isn't explicitly acknowledged. Clarifying the focus and explicitly setting the scene is necessary to improve clarity. For example, it might be easiest to explain that the introduction is focusing on birds that are long-term monogamous, and then introduce the relevant literature in a clear way.

My line-by-line comments are follows:

L42-45: This sentence sets up a false contrast. The time and energy devoted to offspring is costly precisely because it impacts survival and/or future reproduction. If there was no future survival or reproduction (i.e., the species is semelparous), the cost of parental investment wouldn’t matter. Furthermore, I don’t think ‘evolutionarily’ is the right word here – this is describing an individual trade-off. Finally, ‘somehow jeapordizes’ makes it sound like you don’t know how – I suggest removing ‘somehow’.

L50-56: The theory is oversimplified here. Cooperation itself isn’t a solution to sexual conflict because it is evolutionarily unstable. A cooperating pair is always vulnerable to one partner free-loading on the other – there needs to be enforcement for it to be maintained. For long-term monogamous species, this enforcement may come about through the intrinsic benefit of retaining the partner – for example, because there is a high cost to divorce. Cooperation emerges only when (1) parents cannot provide uniparental care and (2) there is a high cost to losing that partner. This concept of ‘partner value’ is reviewed in Griffith 2019.

L59-60: For balance, it is probably worth noting that some studies suggest/show that sexual conflict between the parents can mean that offspring end up receiving less investment in biparental care than would be expected given the sum of the parents’ possible individual contribution, e.g.:

McNamara JM, Houston AI, Barta Z, Osorno JL. 2003. Should young ever be better off with one parent than with two? Behavioral Ecology. 14(3):301–310. doi:10.1093/beheco/14.3.301.

Royle NJ, Hartley IR, Parker GA. 2002. Sexual conflict reduces offspring fitness in zebra finches. Nature. 416(6882):733–736. doi:10.1038/416733a.

L63: I’m not sure the Tyson paper shows a positive relationship between parental body condition and coordination.

L66: It’s unclear what is meant by ‘short time of pair-bonding’

L84: I suggest ‘likely to be /a/ prime driver’ (rather than /the/) – ‘the’ suggests that this is the only driver of parental behaviour, which I suspect is unlikely.

L84-87: I’m not sure what these sentences mean. Is this saying that environmental variables might affect coordination, and might additionally affect coordination differently during different breeding periods? And if this is the case, then we might have different expectations of coordination depending on environmental context?

L88-91: This needs some citations and examples.

L91-92: Why would coordination be more pronounced during chick rearing? There is an example given for incubation but not here.

L93-97: This section really needs some references and examples to support it – it’s quite vague and unspecific. I would also like to see some percentages supporting the statement that most parents don’t coordinate during incubation (I don’t disagree, but you need to substantiate this). Similarly you need to substantiate the claim that tasks are generally more similar between the parents during chick-rearing

L102: Contribution of both parents to care does not necessarily equal coordination. For example, if the female takes on the whole of incubation, and the male takes on the whole (or part) of chick-rearing, they are both contributing to/engaging in care, but no coordination is required.

L108: Is this a general phenomenon? Or are there specific species where this has been tested?

L117: Some authors have indirectly investigated this – e.g. McCully et al 2022 find similar levels of coordination in incubation and brooding:

McCully FR, Weimerskirch H, Cornell SJ, Hatchwell BJ, Cairo M, Patrick SC. 2022. Partner intrinsic characteristics influence foraging trip duration, but not coordination of care in wandering albatrosses Diomedea exulans. Ecology and Evolution. 12(12). doi:10.1002/ece3.9621.

L117-119: This is very vague – please explain fully why they would differ.

L120: What is meant by a runway?

L121: Is there actually evidence for such a short-scale familiarity effect? My understanding is that the familiarity effect moreso relates to the idea that you don’t have to learn a new individual’s behavioural each year (i.e., is more of an annual phenomenon)

L127: I would argue that most species are not genetically monogamous. Extra-pair copulation is extremely common, including in seabirds.

L129-130: It needs to be made clear that cooperation is favoured due to partner value, not similarity in contribution – this is a circular argument.

L152-153: My understanding is that in little auks, the egg is never left unattended. So does this not mean that coordination in this context is a given, i.e. birds are constrained to wait for their partner at the nest? As a general related point, it’s quite unclear what is actually being measured in the study and what the response variables are.

L159: What does ‘habituation to the parental mode’ mean?

L161-162: What is optimized food delivery rate?

L168: Missing the word ‘period’ after incubation (or else remove ‘the’)

Methods

The methods are, in many parts, very difficult to follow. I have done my best to draw a reasonable conclusion about the approaches but there are several sentences that need to be rephrased. Additionally, as discussed above, I am not sure that the methods outlined here actually measure coordination during incubation.

L180: How was an ‘expected event’ determined?

L186-187: I appreciate the intention between only focusing on successful breeders, but there is an interesting question about whether failed breeders coordinate less (which is even mentioned in the discussion), and so I wonder why there is no analysis of this –some sort of simplistic approach that ends up in the Supplementary Materials might be interesting from the perspective of future study, even if it comes with caveats that mean it has to be cautiously interpreted.

L205-208: If I’ve interpreted this correctly, does this mean: in 2019, recordings were taken at fixed intervals following egg laying, and due to the lack of lay dates in 2020, the recordings were taken at fixed calendar dates across the colony (i.e. all birds have the same recording dates)?

L210-212: It would be useful to know how long incubation and chick rearing last for in this species.

If lay date was unknown, how were the recording days determined? It says here ‘days before hatching’ was used, but surely hatching date was unknown if lay date was unknown (at the time of planning the recordings)?

L221: ‘well-adjusted to respective phenology’ – what does this mean? Perhaps ‘timed to hatching phenology’?

L226: I think ‘precision’ rather than ‘accuracy’

L228: Were all videos watched by all observers? Or were they split between observers? If they were split, did you conduct any analysis to look for observer bias?

L229-232: Are there stats to substantiate this, e.g. in the supplementary?

L242-242: Is it possible that the individual could be present at colony for a proportion of this time? Does this matter?

L256: I’m not sure what is meant by constraints. It’s probably enough to say that they represent the main parental activities during incubation.

L258-259: See my earlier comments: if the egg needs to be incubated continuously, how can coordination as established in the introduction exist?

L268-270: If the egg is never left attended, then surely there are very rarely situations when this isn't the case? This seems to be set up as the null hypothesis, but I don't think this is realistic as we know biologically this is always the case, so how can there ever be a situation where coordination is not observed? This approach makes sense for chick-rearing where the requirement for overlap is not so strict, but I'm not sure this approach makes sense for incubation, unless egg neglect is very common.

L279-280: What is the 'observed value'? I’m not sure what is actually being measured here?

L281: Can parents ever do the same activity? If they have to incubate the egg continuously, I'm not sure how one could observe coordination in this context (except maybe in chick rearing). This approach seems to be comparing parental behaviour to something that never happens.

L288: I think this means a correlation between foraging and nest time - this makes more sense and matches previous literature.

L288-293: I’m not sure I follow this.

L295-303: I also don’t follow this.

L305-306: I don't think this is measuring coordination - it sounds like this is just measuring trip duration/nest attendance, because parents always have to be conducting opposite activities

L306-307: Minor point: phase of incubation seems a bit misleading as this is a continuous numeric variable. Perhaps just day of incubation or days until hatching?

L311: What is the distribution? I’m still not clear on what the response variable actually is.

L317: Please explain that verification was ‘by visual inspection of diagnostic plots’ (my interpretation based on the supplementary).

L343: What is a ‘chick-rearing phase’?

L347-351: On one hand, I think the short and long trips for each parent are shuffled in time and compared to the original pattern - this makes sense. But I'm not sure what the 'observed within-pair amount of time...' is.

L357: Mean of what values?

L358-359: I don’t think I understand this, because there doesn’t seem to be a coordination index for incubation.

L366-367: The phase of chick rearing should be explained much earlier on. Why is it split in this way rather than for example using days since hatching (a numeric value), as is done for incubation?

L385: Why not include the phases in the model? Given separate models are fitted, what corrections are applied to account for multiple hypothesis testing?

L386-387: Earlier, phase seems to be days until hatching (numeric), but here it's categorical. In the figures it seems to be numeric. How are these categories chosen and why is this approach used?

Results

L417-419: Perhaps I’ve misunderstood this, but I think it would be more interesting to report how frequently they take trips that are opposite, as opposed to representing this as a percentage time of the recording.

L420: Is 15% a high enough value to claim that coordination is occurring? This seems very low.

L435-436: This seems to be a post-hoc test and not justified in the methods, though I may have missed something. I am a bit concerned about this alternative approach and why the authors have used it – it feels a little like searching for a significant result.

Discussion

I feel the conclusions in the discussion are not always supported by the results, and the inferential chains aren’t always logical. I can see this difficulties in interpreting quite a mixed picture of results, and I wonder if this is partially because coordination is measured during incubation in such a different way to chick-rearing. Generally speaking, the discussion needs to be much better rooted in the published literature. There is very little reference to the wider literature, and the literature cited is a bit too dependent on the authors' previous work (while of course acknowledging that much of this lays the foundation for this study).

L486-487: I don't disagree that the behaviour is flexible, but I don't think this is something that was actually analysed here.

L493-495: I don’t agree with this interpretation – but see my previous comments.

L499-500: Different level is vague and not really what is meant - coordinate better than by chance, I think.

L501-504: This is quite a strong statement when two sentences prior, the authors make the point that there is limited evidence for coordination in chick rearing.

L507: If coordination were integral to this, surely we would expect (and find) quite strong evidence for it?

L511: I don’t think ‘as’ makes sense; the second part of the sentence doesn’t follow from the first part.

L513: ‘As we expected’ – where was this prediction made?

L514-515: I don't see anywhere where it was hypothesised that coordination would depend on the season?

L523-525: This is getting the causation backwards. Hormonal changes are a mechanistic explanation for the behaviour, not an ultimate one. In other words, hormone changes have likely evolved to facilitate the coordination pattern itself, the coordination doesn't emerge because of the hormone changes.

L526: Indeed, I don’t believe you can examine inter-annual differences with just two years, especially with different methodologies.

L535-536: I'm not sure how this conclusion is reached. How is efficiency measured?

L539: Can this be related back to the literature e.g. examples where this is the case?

L543-545: I don't think there is much evidence for this argument, and I don't think it adds anything.

L546: This is still only two years, and chick rearing is short - I doubt the temporal scale makes much difference.

L550-551: But according to the authors coordination is stronger during incubation? So why is coordination not established then (rather than after hatching)?

L560-561: This seems to be the first reference to ‘instability’ – what does this mean? (Same problem on L565).

L570: Not clear what ‘the same’ refers to.

L571-572: This needs to be better substantiated. How would a link between mid-chick rearing and mid incubation tell us anything about quality? What evidence is there for this?.

Reviewer 3

This study is well designed, and the manuscript provides valuable insights into how parental performance in a long-lived, monogamous seabird, emphasizing the flexibility of bi-parental care behavior rather than a fixed approach. It successfully navigates through the complexities of seasonal, stage-dependent, and inter-annual variations in parental coordination, contributing to the broader understanding of bi-parental care in seabirds.

Reviewers' comments:

Reviewer's Responses to Questions

**Comments to the Author**

1. Is the manuscript technically sound, and do the data support the conclusions?

Reviewer #1: Yes

Reviewer #2: Partly

Reviewer #3: Yes

2. Has the statistical analysis been performed appropriately and rigorously? 

Reviewer #1: Yes

Reviewer #2: No

Reviewer #3: N/A

3. Have the authors made all data underlying the findings in their manuscript fully available?

Reviewer #1: Yes

Reviewer #2: Yes

Reviewer #3: No

4. Is the manuscript presented in an intelligible fashion and written in standard English?

Reviewer #1: Yes

Reviewer #2: No

Reviewer #3: Yes

5. Review Comments to the Author

Reviewer #1: It is possible to follow a logic between methods, results and discussion, The authors are careful in the design, even when the Dovekie is broadly studied this work offers different approach.

This work follows the statistical analysis and design according to the characteristics of behavior, reproductive performance, and breeding sites characteristics.

All data is available in the manuscript or supporting materials.

On the line 309 the authors mentioned about inter-annual meteorological and oceanographic variables between years to construct the generalized liner mixed model, but little is mentioned about these variables in the results section, however on the discussion the author mentioned the importance of environmental variables.

Author must review references. Line 50 Parker et al 2002; Line 496 Wojczulanis-Jakuba et al (2018) are not listed. Otherwise on line 704 Jakubas, D., Wojczulanis-Jakubas, K., & Kreft, R. (2008). Sex differences in body condition and hematological parameters in Little Auk Alle alle during the incubation period. Ornis Fennica, 85, 90–97 is on references but not on manuscript.

Reviewer #2: This manuscript explores coordination of parental effort for Dovekies breeding in Svalbard. The study adopts a novel whole-season perspective, whereby the authors consider coordination throughout the entire breeding period and investigate correlations between incubation and chick-rearing, two distinct periods of care underlain by different parental behaviours. While the study poses an intriguing question, and acknowledging the authors' comprehensive approach, I have concerns. The writing is a little raw and difficult to follow, especially in the methods section, where complex approaches lack full explanation. Many conclusions in the discussion seem weakly supported, with unclear logical chains, and insufficient appraisal of the wider literature. However, my major concern is that the authors may not be measuring coordination during incubation accurately.

The defined metric, 'time parents spend on opposite activities,' seems flawed, as continuous egg incubation means parents must nearly always be engaged in opposite activities. As a result, this metric instead seems to just measure respective foraging time/nest attendance, not coordination. Furthermore, I question the randomization procedure, as the null hypothesis represents a pattern of behaviour that fundamentally cannot happen – i.e., two parents behave with no regard to one another, which would leave long periods of egg neglect or overlap at the nest. This is particularly problematic considering that the authors only use successful nests, which presumably are much less likely to have neglected the egg – the only behaviour that would give rise to ‘low’ coordination in this metric. This is a valuable dataset, but I recommend a revised analytical approach – for example, by comparing the nest shifts/trip durations of each parent, as has been done in many other species. Alongside this, I would like to see clearer methods, and a more cautious interpretation of the results.

As a general point, I note quite a few issues with the language usage. In parts, this made it difficult to follow the text and it took me significant effort to parse the meaning. I recognize the complexity of writing in a second language and appreciate the authors' efforts. However, improving this will significantly benefit the overall flow and comprehension and ensure the paper is understood and read by a wide audience. I would strongly suggest that a proficient English speaker is involved in the revision process to ensure that the intended meaning is effectively conveyed, particularly as PLOS ONE does not offer copy editing. I hope that the Editor can facilitate this.

Below I have outlined my thoughts in more detail. I hope this helps the authors improve their manuscript, and I look forward to seeing an updated version.

Introduction

The introduction contains the required theory to understand the background of the study, which is well justified. However, I’m afraid I feel it needs work. The presentation of theory is quite disorganised, with several vague explanations and examples. Additional citations, concrete examples, and supporting details are needed in various parts. Similarly, the study's description at the end is unclear, especially regarding the definition of coordination in this context.

One issue is that the context and focus of the introduction is not well established – it is unclear whether it focuses on all caregiving animals, species with biparental care, or just long-term monogamous parents. The entire introduction seems bird-centric, yet this isn't explicitly acknowledged. Clarifying the focus and explicitly setting the scene is necessary to improve clarity. For example, it might be easiest to explain that the introduction is focusing on birds that are long-term monogamous, and then introduce the relevant literature in a clear way.

My line-by-line comments are follows:

L42-45: This sentence sets up a false contrast. The time and energy devoted to offspring is costly precisely because it impacts survival and/or future reproduction. If there was no future survival or reproduction (i.e., the species is semelparous), the cost of parental investment wouldn’t matter. Furthermore, I don’t think ‘evolutionarily’ is the right word here – this is describing an individual trade-off. Finally, ‘somehow jeapordizes’ makes it sound like you don’t know how – I suggest removing ‘somehow’.

L50-56: The theory is oversimplified here. Cooperation itself isn’t a solution to sexual conflict because it is evolutionarily unstable. A cooperating pair is always vulnerable to one partner free-loading on the other – there needs to be enforcement for it to be maintained. For long-term monogamous species, this enforcement may come about through the intrinsic benefit of retaining the partner – for example, because there is a high cost to divorce. Cooperation emerges only when (1) parents cannot provide uniparental care and (2) there is a high cost to losing that partner. This concept of ‘partner value’ is reviewed in Griffith 2019.

L59-60: For balance, it is probably worth noting that some studies suggest/show that sexual conflict between the parents can mean that offspring end up receiving less investment in biparental care than would be expected given the sum of the parents’ possible individual contribution, e.g.:

McNamara JM, Houston AI, Barta Z, Osorno JL. 2003. Should young ever be better off with one parent than with two? Behavioral Ecology. 14(3):301–310. doi:10.1093/beheco/14.3.301.

Royle NJ, Hartley IR, Parker GA. 2002. Sexual conflict reduces offspring fitness in zebra finches. Nature. 416(6882):733–736. doi:10.1038/416733a.

L63: I’m not sure the Tyson paper shows a positive relationship between parental body condition and coordination.

L66: It’s unclear what is meant by ‘short time of pair-bonding’

L84: I suggest ‘likely to be /a/ prime driver’ (rather than /the/) – ‘the’ suggests that this is the only driver of parental behaviour, which I suspect is unlikely.

L84-87: I’m not sure what these sentences mean. Is this saying that environmental variables might affect coordination, and might additionally affect coordination differently during different breeding periods? And if this is the case, then we might have different expectations of coordination depending on environmental context?

L88-91: This needs some citations and examples.

L91-92: Why would coordination be more pronounced during chick rearing? There is an example given for incubation but not here.

L93-97: This section really needs some references and examples to support it – it’s quite vague and unspecific. I would also like to see some percentages supporting the statement that most parents don’t coordinate during incubation (I don’t disagree, but you need to substantiate this). Similarly you need to substantiate the claim that tasks are generally more similar between the parents during chick-rearing

L102: Contribution of both parents to care does not necessarily equal coordination. For example, if the female takes on the whole of incubation, and the male takes on the whole (or part) of chick-rearing, they are both contributing to/engaging in care, but no coordination is required.

L108: Is this a general phenomenon? Or are there specific species where this has been tested?

L117: Some authors have indirectly investigated this – e.g. McCully et al 2022 find similar levels of coordination in incubation and brooding:

McCully FR, Weimerskirch H, Cornell SJ, Hatchwell BJ, Cairo M, Patrick SC. 2022. Partner intrinsic characteristics influence foraging trip duration, but not coordination of care in wandering albatrosses Diomedea exulans. Ecology and Evolution. 12(12). doi:10.1002/ece3.9621.

L117-119: This is very vague – please explain fully why they would differ.

L120: What is meant by a runway?

L121: Is there actually evidence for such a short-scale familiarity effect? My understanding is that the familiarity effect moreso relates to the idea that you don’t have to learn a new individual’s behavioural each year (i.e., is more of an annual phenomenon)

L127: I would argue that most species are not genetically monogamous. Extra-pair copulation is extremely common, including in seabirds.

L129-130: It needs to be made clear that cooperation is favoured due to partner value, not similarity in contribution – this is a circular argument.

L152-153: My understanding is that in little auks, the egg is never left unattended. So does this not mean that coordination in this context is a given, i.e. birds are constrained to wait for their partner at the nest? As a general related point, it’s quite unclear what is actually being measured in the study and what the response variables are.

L159: What does ‘habituation to the parental mode’ mean?

L161-162: What is optimized food delivery rate?

L168: Missing the word ‘period’ after incubation (or else remove ‘the’)

Methods

The methods are, in many parts, very difficult to follow. I have done my best to draw a reasonable conclusion about the approaches but there are several sentences that need to be rephrased. Additionally, as discussed above, I am not sure that the methods outlined here actually measure coordination during incubation.

L180: How was an ‘expected event’ determined?

L186-187: I appreciate the intention between only focusing on successful breeders, but there is an interesting question about whether failed breeders coordinate less (which is even mentioned in the discussion), and so I wonder why there is no analysis of this –some sort of simplistic approach that ends up in the Supplementary Materials might be interesting from the perspective of future study, even if it comes with caveats that mean it has to be cautiously interpreted.

L205-208: If I’ve interpreted this correctly, does this mean: in 2019, recordings were taken at fixed intervals following egg laying, and due to the lack of lay dates in 2020, the recordings were taken at fixed calendar dates across the colony (i.e. all birds have the same recording dates)?

L210-212: It would be useful to know how long incubation and chick rearing last for in this species.

If lay date was unknown, how were the recording days determined? It says here ‘days before hatching’ was used, but surely hatching date was unknown if lay date was unknown (at the time of planning the recordings)?

L221: ‘well-adjusted to respective phenology’ – what does this mean? Perhaps ‘timed to hatching phenology’?

L226: I think ‘precision’ rather than ‘accuracy’

L228: Were all videos watched by all observers? Or were they split between observers? If they were split, did you conduct any analysis to look for observer bias?

L229-232: Are there stats to substantiate this, e.g. in the supplementary?

L242-242: Is it possible that the individual could be present at colony for a proportion of this time? Does this matter?

L256: I’m not sure what is meant by constraints. It’s probably enough to say that they represent the main parental activities during incubation.

L258-259: See my earlier comments: if the egg needs to be incubated continuously, how can coordination as established in the introduction exist?

L268-270: If the egg is never left attended, then surely there are very rarely situations when this isn't the case? This seems to be set up as the null hypothesis, but I don't think this is realistic as we know biologically this is always the case, so how can there ever be a situation where coordination is not observed? This approach makes sense for chick-rearing where the requirement for overlap is not so strict, but I'm not sure this approach makes sense for incubation, unless egg neglect is very common.

L279-280: What is the 'observed value'? I’m not sure what is actually being measured here?

L281: Can parents ever do the same activity? If they have to incubate the egg continuously, I'm not sure how one could observe coordination in this context (except maybe in chick rearing). This approach seems to be comparing parental behaviour to something that never happens.

L288: I think this means a correlation between foraging and nest time - this makes more sense and matches previous literature.

L288-293: I’m not sure I follow this.

L295-303: I also don’t follow this.

L305-306: I don't think this is measuring coordination - it sounds like this is just measuring trip duration/nest attendance, because parents always have to be conducting opposite activities

L306-307: Minor point: phase of incubation seems a bit misleading as this is a continuous numeric variable. Perhaps just day of incubation or days until hatching?

L311: What is the distribution? I’m still not clear on what the response variable actually is.

L317: Please explain that verification was ‘by visual inspection of diagnostic plots’ (my interpretation based on the supplementary).

L343: What is a ‘chick-rearing phase’?

L347-351: On one hand, I think the short and long trips for each parent are shuffled in time and compared to the original pattern - this makes sense. But I'm not sure what the 'observed within-pair amount of time...' is.

L357: Mean of what values?

L358-359: I don’t think I understand this, because there doesn’t seem to be a coordination index for incubation.

L366-367: The phase of chick rearing should be explained much earlier on. Why is it split in this way rather than for example using days since hatching (a numeric value), as is done for incubation?

L385: Why not include the phases in the model? Given separate models are fitted, what corrections are applied to account for multiple hypothesis testing?

L386-387: Earlier, phase seems to be days until hatching (numeric), but here it's categorical. In the figures it seems to be numeric. How are these categories chosen and why is this approach used?

Results

L417-419: Perhaps I’ve misunderstood this, but I think it would be more interesting to report how frequently they take trips that are opposite, as opposed to representing this as a percentage time of the recording.

L420: Is 15% a high enough value to claim that coordination is occurring? This seems very low.

L435-436: This seems to be a post-hoc test and not justified in the methods, though I may have missed something. I am a bit concerned about this alternative approach and why the authors have used it – it feels a little like searching for a significant result.

Discussion

I feel the conclusions in the discussion are not always supported by the results, and the inferential chains aren’t always logical. I can see this difficulties in interpreting quite a mixed picture of results, and I wonder if this is partially because coordination is measured during incubation in such a different way to chick-rearing. Generally speaking, the discussion needs to be much better rooted in the published literature. There is very little reference to the wider literature, and the literature cited is a bit too dependent on the authors' previous work (while of course acknowledging that much of this lays the foundation for this study).

L486-487: I don't disagree that the behaviour is flexible, but I don't think this is something that was actually analysed here.

L493-495: I don’t agree with this interpretation – but see my previous comments.

L499-500: Different level is vague and not really what is meant - coordinate better than by chance, I think.

L501-504: This is quite a strong statement when two sentences prior, the authors make the point that there is limited evidence for coordination in chick rearing.

L507: If coordination were integral to this, surely we would expect (and find) quite strong evidence for it?

L511: I don’t think ‘as’ makes sense; the second part of the sentence doesn’t follow from the first part.

L513: ‘As we expected’ – where was this prediction made?

L514-515: I don't see anywhere where it was hypothesised that coordination would depend on the season?

L523-525: This is getting the causation backwards. Hormonal changes are a mechanistic explanation for the behaviour, not an ultimate one. In other words, hormone changes have likely evolved to facilitate the coordination pattern itself, the coordination doesn't emerge because of the hormone changes.

L526: Indeed, I don’t believe you can examine inter-annual differences with just two years, especially with different methodologies.

L535-536: I'm not sure how this conclusion is reached. How is efficiency measured?

L539: Can this be related back to the literature e.g. examples where this is the case?

L543-545: I don't think there is much evidence for this argument, and I don't think it adds anything.

L546: This is still only two years, and chick rearing is short - I doubt the temporal scale makes much difference.

L550-551: But according to the authors coordination is stronger during incubation? So why is coordination not established then (rather than after hatching)?

L560-561: This seems to be the first reference to ‘instability’ – what does this mean? (Same problem on L565).

L570: Not clear what ‘the same’ refers to.

L571-572: This needs to be better substantiated. How would a link between mid-chick rearing and mid incubation tell us anything about quality? What evidence is there for this?

Reviewer #3: This study is well designed, and the manuscript provides valuable insights into how parental performance in a long-lived, monogamous seabird, emphasizing the flexibility of bi-parental care behavior rather than a fixed approach. It successfully navigates through the complexities of seasonal, stage-dependent, and inter-annual variations in parental coordination, contributing to the broader understanding of bi-parental care in seabirds.

6. PLOS authors have the option to publish the peer review history of their article (what does this mean?). If published, this will include your full peer review and any attached files.

Reviewer #1: **Yes: **María Félix-Lizárraga

Reviewer #2: No

Reviewer #3: No

---

## [Author Response · Author response to Decision Letter 0]

13 Mar 2024

Please find below a copy of the Decision letter, with specific replies to comments. This can also be found in the provided document OTHER/Response to Review, for easier identification of questions and replies using a color code.

Dear Dr. Grissot,

Thank you for submitting your manuscript to PLOS ONE. After careful consideration, we feel that it has merit but does not fully meet PLOS ONE’s publication criteria as it currently stands. Therefore, we invite you to submit a revised version of the manuscript that addresses the points raised during the review process.

Dear Dr., Antoine Grissot

Thank you for submitting your manuscript to PLOS ONE. After careful consideration, we have decided that your manuscript needs Major Revision.

Kind regards,

Prof. Lamiaa Mostafa Radwan, Ph.D.

Academic Editor

PLOS ONE

Editor Comments:

1- The manuscript needs Editing language

2- Material and methods need more clarity

3- Discussing results requires citing references that explain those results

Dear PlosOne editor,

Thank you very much for your consideration of our work and your positive and constructive comments. Please find in the revision our updated version of the manuscript, that we hope will satisfy the comments raised by you and the three reviewers. Concerning your three specific comments:

1- We carefully edited the manuscript with special care for language.

2- Following this and mainly reviewer 2 comments, we did our best to explain better and clarify everything in the Method section

3- We improved the discussion and added needed references.

When providing Line numbers, we refer to the document with TRACK CHANGE activated.

Reviewer1

It is possible to follow a logic between methods, results and discussion, The authors are careful in the design, even when the Dovekie is broadly studied this work offers different approach.

This work follows the statistical analysis and design according to the characteristics of behavior, reproductive performance, and breeding sites characteristics.

All data is available in the manuscript or supporting materials.

Dear Reviewer1, thank you very much for your work on our manuscript and your comments.

On the line 309 the authors mentioned about inter-annual meteorological and oceanographic variables between years to construct the generalized liner mixed model, but little is mentioned about these variables in the results section, however on the discussion the author mentioned the importance of environmental variables.

REPLY : Sorry if that wasn’t clear in the method section, but only year was included as a variable in the model, not the meteorological and oceanographic variables. We believe including the year can be useful to take into consideration variation in environmental conditions, without including them directly, as this would complicate the model a lot and is not the main purpose of the study. We also included importance of environmental variables in the discussion, as it is known from the literature cited, but once again was not the main purpose here.

Author must review references. Line 50 Parker et al 2002; Line 496 Wojczulanis-Jakuba et al (2018) are not listed. Otherwise on line 704 Jakubas, D., Wojczulanis-Jakubas, K., & Kreft, R. (2008). Sex differences in body condition and hematological parameters in Little Auk Alle alle during the incubation period. Ornis Fennica, 85, 90–97 is on references but not on manuscript.

REPLY : References were carefully checked in the revised version. Regarding Wojczulanis-Jakubas et al. (2018), this was just an omission of the “2018a”. The two other mentioned occurrences were respectively added for Parker et al 2002 and removed for Jakubas et al 2008.

Reviewer2

This manuscript explores coordination of parental effort for Dovekies breeding in Svalbard. The study adopts a novel whole-season perspective, whereby the authors consider coordination throughout the entire breeding period and investigate correlations between incubation and chick-rearing, two distinct periods of care underlain by different parental behaviours. While the study poses an intriguing question, and acknowledging the authors' comprehensive approach, I have concerns. The writing is a little raw and difficult to follow, especially in the methods section, where complex approaches lack full explanation. Many conclusions in the discussion seem weakly supported, with unclear logical chains, and insufficient appraisal of the wider literature. However, my major concern is that the authors may not be measuring coordination during incubation accurately.

REPLY : We thank the reviewer for the comments on our manuscript, and hope we addressed present and detailed concerns in a satisfying manner

The defined metric, 'time parents spend on opposite activities,' seems flawed, as continuous egg incubation means parents must nearly always be engaged in opposite activities. As a result, this metric instead seems to just measure respective foraging time/nest attendance, not coordination. Furthermore, I question the randomization procedure, as the null hypothesis represents a pattern of behaviour that fundamentally cannot happen – i.e., two parents behave with no regard to one another, which would leave long periods of egg neglect or overlap at the nest. This is particularly problematic considering that the authors only use successful nests, which presumably are much less likely to have neglected the egg – the only behaviour that would give rise to ‘low’ coordination in this metric. This is a valuable dataset, but I recommend a revised analytical approach – for example, by comparing the nest shifts/trip durations of each parent, as has been done in many other species. Alongside this, I would like to see clearer methods, and a more cautious interpretation of the results.

REPLY : We understand the reviewer’s concerns about methodology, and did our best to explain in a clearer way all the methods used. However, we disagree concerning the alleged flaws in the coordination metrics and the randomisation method.

First, we added in the text (lines 273-275) some nuance about the constant need for incubation. Indeed, from previous work and the present study, we know there is some amount of tolerance for the egg not to be constantly incubated. Thermoregulation of the egg depends on incubating behavior of parents, but if left unattended, temperature of the egg does not drop below a critical threshold straight away and therefore can be left alone for some time. While processing the videos we could notice that sometimes an incubating parent would leave the nest before the other parent would return, and this, if happening rarely, didn’t seem to affect hatching success.

Additionally, the time performing opposite activities does not “just measure respective foraging time/nest attendance” because of the presence of the third category (“colony”), that can reduce the performance of opposite activities. Some individuals spent a considerate amount of time in the colony, either alone or with the partner, and even if the function of this behaviour is not very well known, it could serve as an alternative resting for a non-incubating individual (staying in colony instead of performing a foraging trip while partner incubates the egg), or as a bonding time between parents. Additionally, the two other categories, “nest” and “foraging”, perfectly represent the needs of respectively the egg and the parents, but only when performed simultaneously do they represent an optimisation of the time allocation of both parents in accord with each other. Thus, with the extent of tolerance for egg being left alone, and other activities being possibly performed, the performance of opposite activities really represents coordination between the two parents. This is also supported by the fact that there is actual variability in the dataset of this study in the time that parent spent performing opposite activities, so parents are not nearly always engaged in opposite activities as suggested by the reviewer. Indeed, even though all the pairs included in the study were successful, amount of time performing opposite activities is not same for all the pairs (mean = 88%, [min – max] : [48 – 99] %).

Regarding the randomisation process, the idea is to test whether the actual patterns from both parents are different from what is expected by chance. So one of the chance possibilities is what the reviewer suggest (where two parents behave completely with no regard to one another) and we agree that this pattern is not compatible with successful hatching, however our null hypothesis also encompasses a whole range of different situations in between. Therefore, we believe that testing the significance of this observed pattern compared to what could arise by chance is still valuable in itself. 

As a general point, I note quite a few issues with the language usage. In parts, this made it difficult to follow the text and it took me significant effort to parse the meaning. I recognize the complexity of writing in a second language and appreciate the authors' efforts. However, improving this will significantly benefit the overall flow and comprehension and ensure the paper is understood and read by a wide audience. I would strongly suggest that a proficient English speaker is involved in the revision process to ensure that the intended meaning is effectively conveyed, particularly as PLOS ONE does not offer copy editing. I hope that the Editor can facilitate this.

REPLY : Thank you for acknowledging the efforts and complexity of writing in a second language. We did our best to correct the present manuscript with language in mind, and once every scientific issues are resolved and manuscript is deemed satisfying by the editor, we will submit the last version of the text to a native English speaker who is also a specialist in ecology, for proof reading and improve the language.

Below I have outlined my thoughts in more detail. I hope this helps the authors improve their manuscript, and I look forward to seeing an updated version.

Introduction

The introduction contains the required theory to understand the background of the study, which is well justified. However, I’m afraid I feel it needs work. The presentation of theory is quite disorganised, with several vague explanations and examples. Additional citations, concrete examples, and supporting details are needed in various parts. Similarly, the study's description at the end is unclear, especially regarding the definition of coordination in this context.

One issue is that the context and focus of the introduction is not well established – it is unclear whether it focuses on all caregiving animals, species with biparental care, or just long-term monogamous parents. The entire introduction seems bird-centric, yet this isn't explicitly acknowledged. Clarifying the focus and explicitly setting the scene is necessary to improve clarity. For example, it might be easiest to explain that the introduction is focusing on birds that are long-term monogamous, and then introduce the relevant literature in a clear way.

REPLY : We tried to make it clearer in the introduction that we are mainly focusing on bird species with bi-parental care. Given that PlosOne is a Journal with a wide range of scientific interests, we started the introduction quite widely on parental care and narrowed it down to the specific case of seabirds with long-term pair bond and biparental care.

My line-by-line comments are follows:

L42-45: This sentence sets up a false contrast. The time and energy devoted to offspring is costly precisely because it impacts survival and/or future reproduction. If there was no future survival or reproduction (i.e., the species is semelparous), the cost of parental investment wouldn’t matter. Furthermore, I don’t think ‘evolutionarily’ is the right word here – this is describing an individual trade-off. Finally, ‘somehow jeapordizes’ makes it sound like you don’t know how – I suggest removing ‘somehow’.

REPLY : We rephrased the sentence, see lines 42-45.

L50-56: The theory is oversimplified here. Cooperation itself isn’t a solution to sexual conflict because it is evolutionarily unstable. A cooperating pair is always vulnerable to one partner free-loading on the other – there needs to be enforcement for it to be maintained. For long-term monogamous species, this enforcement may come about through the intrinsic benefit of retaining the partner – for example, because there is a high cost to divorce. Cooperation emerges only when (1) parents cannot provide uniparental care and (2) there is a high cost to losing that partner. This concept of ‘partner value’ is reviewed in Griffith 2019.

REPLY : We removed mention of “solution” (see line 55) but nonetheless believe that coordination allows some mitigation of the conflict, through the two mechanisms mentioned by the reviewer, that provide stability.

L59-60: For balance, it is probably worth noting that some studies suggest/show that sexual conflict between the parents can mean that offspring end up receiving less investment in biparental care than would be expected given the sum of the parents’ possible individual contribution, e.g.:

McNamara JM, Houston AI, Barta Z, Osorno JL. 2003. Should young ever be better off with one parent than with two? Behavioral Ecology. 14(3):301–310. doi:10.1093/beheco/14.3.301.

Royle NJ, Hartley IR, Parker GA. 2002. Sexual conflict reduces offspring fitness in zebra finches. Nature. 416(6882):733–736. doi:10.1038/416733a.

REPLY : Text was modified and suggested references added, see line 61.

L63: I’m not sure the Tyson paper shows a positive relationship between parental body condition and coordination.

REPLY : The discussion of the Tyson paper briefly mentions parents could have direct “benefit by coordinating, for instance by determining which partner is in greater need of a long, self-maintenance foraging trip”.

L66: It’s unclear what is meant by ‘short time of pair-bonding’

REPLY : It means that the familiarity effect can reduce the time dedicated to re-establishing the pair bond after winter separation.

L84: I suggest ‘likely to be /a/ prime driver’ (rather than /the/) – ‘the’ suggests that this is the only driver of parental behaviour, which I suspect is unlikely.

REPLY : Corrected

L84-87: I’m not sure what these sentences mean. Is this saying that environmental variables might affect coordination, and might additionally affect coordination differently during different breeding periods? And if this is the case, then we might have different expectations of coordination depending on environmental context?

REPLY : As explained by the following sentences (lines 90-96), it is indeed meaning that environmental conditions could influence coordination, and could do so differently for different breeding periods.

L88-91: This needs some citations and examples.

REPLY : This paragraph is used to introduce a very important part of the study that is actually missing from literature. There is no study, in our knowledge, that investigated coordination at both incubation and chick rearing period. Therefore, we present here hypotheses of what is likely to happen depending on differences in species’ ecology.

L91-92: Why would coordination be more pronounced during chick rearing? There is an example given for incubation but not here.

REPLY : As mentioned above, no other study investigated the coordination in both breeding periods, and therefore this sentence highlights that we do not have full understanding of what happens, but we can expect differences between species.

L93-97: This section really needs some references and examples to support it – it’s quite vague and unspecific. I would also like to see some percentages supporting the statement that most parents don’t coordinate during incubation (I don’t disagree, but you need to substantiate this). Similarly you need to substantiate the claim that tasks are generally more similar between the parents during chick-rearing

REPLY : We couldn’t find accurate percentages, but Cockburn (2006) describes quite well all the claims of this section.

L102: Contribution of both parent

---

## [Decision Letter · Decision Letter 1]

3 Apr 2024

PONE-D-23-40200R1Coordination of parental performance is breeding phase-dependent in the Dovekie (Alle alle), a pelagic Arctic seabirdPLOS ONE

Dear Dr. Grissot,

Thank you for submitting your manuscript to PLOS ONE. After careful consideration, we feel that it has merit but does not fully meet PLOS ONE’s publication criteria as it currently stands. Therefore, we invite you to submit a revised version of the manuscript that addresses the points raised during the review process.

We look forward to receiving your revised manuscript.

Kind regards,

Lamiaa Mostafa Radwan, Ph.D.

Academic Editor

PLOS ONE

Journal Requirements:

**Additional Editor Comments:**

Dear Dr., Antoine Grissot

Thank you for submitting your manuscript to PLOS ONE. After careful consideration, we have decided that your manuscript needs Minor Revision.

Comment Editor:

It is necessary to make all amendments and corrections requested by Reviewer 2

Kind regards,

Prof. Lamiaa Mostafa Radwan, Ph.D.

Academic Editor

PLOS ONE

Reviewer1

Accept

Reviewer2

Many thanks to the authors for their efforts in revising their manuscript. I appreciate that they have implemented some of my feedback, as well as their additional clarification regarding the coordination metrics. However, while the authors have addressed my comments in their rebuttal letter, and for the most part done so very well and clearly, I found that these changes have not consistently translated into edits to the manuscript itself. Consequently, many of my comments below are identical to those provided in the previous review, with explicit requests that the well-explained responses provided by the authors are integrated directly into the manuscript text. Rectifying this should involve little effort beyond incorporating the content from the rebuttal directly into the manuscript, as is standard practice during revisions.

My concerns about the analyses have been somewhat alleviated but there is one part I still do not understand – the authors make the argument that their coordination analyses work on the basis that birds can engage in three different behaviours: nest, colony, and foraging. Yet, in the rebuttal, they state:

‘Individuals when present in the colony usually spend most of their time in the surroundings of their nest. Furthermore when they leave the frame and come back with a full gular pouch, we know a foraging trip was performed. There is a possibility they spend a little time in the colony before actually departing for the foraging trip, but assume that it is negligible.’

If colony time is assumed to be negligible, I don’t understand how the rest of the analyses follow? This is very likely a misunderstanding on my part, but it has come from unclear explanations. If the authors address this, please ensure this is reflected in the manuscript rather than just the response.

Finally - this is more of a point to the editor - as the authors acknowledge, the manuscript still requires proof reading, but I note that they have made arrangements to do this for their next revision. I have therefore not addressed any typographical or grammatical errors.

My relevant line-by-line comments are below, and refer to the ‘tracked changes’ version of the manuscript.

Introduction

L54: Possibly debatable whether 10 years ago is ‘recently’, I’d suggest removing this word.

L61: These papers show *lower* offspring fitness due to biparental care, and so are improperly cited. The point of my original comment is that while biparental care may improve offspring fitness, this is not guaranteed. McNamara demonstrates this theoretically; Royle shows this experimentally in zebra finches. All this requires is something like ‘(*but see* Royle et al 2003, McNamara et al 2003)’ but I also suggest reading the papers as they are important pieces of theory.

L68: Suggest ‘reduced time investment in pair bonding’ rather than ‘short time of pair-bonding’ for clarity.

L87-88: It is still necessary to explain why environment is likely to be important i.e. give some examples of why/when environment impacts coordination and why it differs between stages. E.g. food availability might differ seasonally, which could be more important in chick rearing when energetic constraints are higher. I appreciate that the authors are investigating novel questions, but they are not plucked from thin air, it’s important to evidence the hypotheses.

L90: Similarly, there are examples in the literature that could be provided here. Even if other authors haven’t specifically compared breeding phases, there are plenty of other studies that could be used to support this e.g. those showing differences between breeding phases in trip duration, in parental behaviour, in specific constraints. ‘May thus be specific…’ is too vague.

L96: This is the sunk cost fallacy – animals never make decisions based on prior investment alone. If a breeding attempt is doomed to fail parents will end the attempt regardless of how much prior investment they have made. Rather, to make this argument one could discuss that it is more costly to start afresh than to continue. But this is a subtle difference. Please see Dawkins & Carlisle’s original explanation: DAWKINS, R., CARLISLE, T. Parental investment, mate desertion and a fallacy. Nature 262, 131–133 (1976). https://doi.org/10.1038/262131a0

This sentence should therefore be removed. I actually think your next few sentences (discussing the idea that many birds don’t coordinate incubation but do coordinate chick rearing) illustrates the idea that coordination might be more pronounced during chick rearing just fine. You could also discuss the increased energetic costs of chick rearing vs incubation and the increased constraints associated with dividing self-care and chick care, for which the literature is full of examples.

L122-123: It’s not enough to just discuss ‘patterns’ – please explain what you mean and give examples (e.g. lots of seabirds where foraging trips are longer during incubation than chick-rearing). As pointed out in the rebuttal, PLOS is a broad journal, so it is important to explain what this means behaviourally.

L125: ‘Runway’ is not a term I have encountered in the literature, and I doubt I’m alone – I suggest explaining as it was in the rebuttal letter i.e. parents may need time to synchronise to each other’s behaviour prior to breeding.

L126: Given there is no evidence (based on the rebuttal letter) I suggest removing; ‘familiarity’ is poorly defined in the literature anyway.

L157: It is essential to state that eggs can withstand neglect, and how long for. If eggs could only survive 10 minutes of neglect, then my original point that successful breeders are constrained to apparently coordinate would still stand. This is crucial to the rest of the analyses so needs to be stated explicitly and clearly.

L166: I still don’t understand what is meant by ‘habituation to the parental mode’ even with the rebuttal letter. This needs explanation – perhaps ‘increased within-pair synchronicity’?

L244: Please include explanation from rebuttal letter for why observer effects are not a concern. In the rebuttal it states ‘we made sure new observer would not differ’ – how, if there were no stats? Worth outlining in supplementary.

L254: Please include explanation from the rebuttal letter.

L271: See my main comments above about whether colony behaviour is negligible or not.

L272: I don’t think ‘and their constraints’ adds anything here, and it’s confusing. I suggest removing it.

L296: Please include explanation from the rebuttal letter.

L323-324: ‘Phase’ is used to mean something different for incubation and chick rearing. This is really confusing as they are

different metrics (continuous numerical vs categorical) and is misleading. Please rephrase.

L328: What is the distribution of the response variable, please state.

L360: State that the two chick rearing phases are ‘early’ and ‘mid’ (perhaps in brackets). There is a lot of reliance on readers remembering quite specific pieces of information, please restate these sorts of things when relevant to make it easier.

L364-368: This is still a very difficult sentence to follow and very long. Please rephrase.

L376-377: My confusion here arose because the phrasing suggests that there was a coordination index for both incubation

and chick rearing (‘In contrast to the incubation period, we decided to use for the chick rearing period the coordination index’), which I understand from the rebuttal letter is not correct. Please rephrase.

L384: Please include explanation from the rebuttal letter.

L506: I suggest some caveat e.g. behaviour is flexible between breeding stages, because you don’t show intra-individual flexibility and therefore can’t technically refute the ‘sealed bid’ model (e.g. parents could submit a fixed sealed bid for incubation and chick rearing separately, which doesn’t change – this is unlikely but we can’t say with certainty).

L544-545: In my original review, I did not mean that hormones don't play a role. My point is that in the discussion, you seem to be asking evolutionary/ultimate questions – i.e. what has driven the evolution of coordinated incubation? For example, you point out that coordination may evolve because it favours embryo development or because it helps to protect chicks prior to thermal independence.

Yes, hormones are likely to be a mechanism by which coordination is achieved – but *why* has this mechanism evolved? It is suggested that hormones explain why birds become more devoted to incubation over time – mechanistically this might be true, but it doesn't answer the 'why' question. Many birds fly south for winter - this could be driven by some hormonally mediated instinct, but that doesn't tell us why they do it in the first place.

It's fine to say that hormonal changes regulate behaviour, but I would put this somewhere else in the discussion and make clear that this is a mechanistic question. In fact, a better option might be to put this in the introduction to help show that levels of coordination change over time. E.g. something along the lines of: prolactin increases over the course of incubation, suggesting changes in parental care behaviour over time - but whether this increases parents’ propensity to incubate, and therefore their level of coordination, is unknown.

L565-567: Please include explanation from the rebuttal letter.

L572-573: It seems surprising to expect coordination to ‘reset’ after chick-rearing as outlined in the rebuttal; this needs further explanation (in the manuscript).

L582: I think perhaps ‘differences’ would make more sense– instability is still not clear.

L594: I suggest ‘reflect’ as ‘express’ suggests an active choice.

L595-596: Please include explanation from the rebuttal letter.

Reviewer3

Accept

Reviewers' comments:

Reviewer's Responses to Questions

**Comments to the Author**

1. If the authors have adequately addressed your comments raised in a previous round of review and you feel that this manuscript is now acceptable for publication, you may indicate that here to bypass the “Comments to the Author” section, enter your conflict of interest statement in the “Confidential to Editor” section, and submit your "Accept" recommendation.

Reviewer #1: All comments have been addressed

Reviewer #2: (No Response)

Reviewer #3: (No Response)

2. Is the manuscript technically sound, and do the data support the conclusions?

Reviewer #1: Yes

Reviewer #2: Yes

Reviewer #3: (No Response)

3. Has the statistical analysis been performed appropriately and rigorously? 

Reviewer #1: Yes

Reviewer #2: Yes

Reviewer #3: (No Response)

4. Have the authors made all data underlying the findings in their manuscript fully available?

Reviewer #1: Yes

Reviewer #2: Yes

Reviewer #3: (No Response)

5. Is the manuscript presented in an intelligible fashion and written in standard English?

Reviewer #1: Yes

Reviewer #2: No

Reviewer #3: (No Response)

6. Review Comments to the Author

Reviewer #1: The authors address all comments one by one, and on those who disagree with the reviewers their argument was strong enough. The manuscript links the methods with the results and discussions. The data is fully available.

Reviewer #2: Many thanks to the authors for their efforts in revising their manuscript. I appreciate that they have implemented some of my feedback, as well as their additional clarification regarding the coordination metrics. However, while the authors have addressed my comments in their rebuttal letter, and for the most part done so very well and clearly, I found that these changes have not consistently translated into edits to the manuscript itself. Consequently, many of my comments below are identical to those provided in the previous review, with explicit requests that the well-explained responses provided by the authors are integrated directly into the manuscript text. Rectifying this should involve little effort beyond incorporating the content from the rebuttal directly into the manuscript, as is standard practice during revisions.

My concerns about the analyses have been somewhat alleviated but there is one part I still do not understand – the authors make the argument that their coordination analyses work on the basis that birds can engage in three different behaviours: nest, colony, and foraging. Yet, in the rebuttal, they state:

‘Individuals when present in the colony usually spend most of their time in the surroundings of their nest. Furthermore when they leave the frame and come back with a full gular pouch, we know a foraging trip was performed. There is a possibility they spend a little time in the colony before actually departing for the foraging trip, but assume that it is negligible.’

If colony time is assumed to be negligible, I don’t understand how the rest of the analyses follow? This is very likely a misunderstanding on my part, but it has come from unclear explanations. If the authors address this, please ensure this is reflected in the manuscript rather than just the response.

Finally - this is more of a point to the editor - as the authors acknowledge, the manuscript still requires proof reading, but I note that they have made arrangements to do this for their next revision. I have therefore not addressed any typographical or grammatical errors.

My relevant line-by-line comments are below, and refer to the ‘tracked changes’ version of the manuscript.

Introduction

L54: Possibly debatable whether 10 years ago is ‘recently’, I’d suggest removing this word.

L61: These papers show *lower* offspring fitness due to biparental care, and so are improperly cited. The point of my original comment is that while biparental care may improve offspring fitness, this is not guaranteed. McNamara demonstrates this theoretically; Royle shows this experimentally in zebra finches. All this requires is something like ‘(*but see* Royle et al 2003, McNamara et al 2003)’ but I also suggest reading the papers as they are important pieces of theory.

L68: Suggest ‘reduced time investment in pair bonding’ rather than ‘short time of pair-bonding’ for clarity.

L87-88: It is still necessary to explain why environment is likely to be important i.e. give some examples of why/when environment impacts coordination and why it differs between stages. E.g. food availability might differ seasonally, which could be more important in chick rearing when energetic constraints are higher. I appreciate that the authors are investigating novel questions, but they are not plucked from thin air, it’s important to evidence the hypotheses.

L90: Similarly, there are examples in the literature that could be provided here. Even if other authors haven’t specifically compared breeding phases, there are plenty of other studies that could be used to support this e.g. those showing differences between breeding phases in trip duration, in parental behaviour, in specific constraints. ‘May thus be specific…’ is too vague.

L96: This is the sunk cost fallacy – animals never make decisions based on prior investment alone. If a breeding attempt is doomed to fail parents will end the attempt regardless of how much prior investment they have made. Rather, to make this argument one could discuss that it is more costly to start afresh than to continue. But this is a subtle difference. Please see Dawkins & Carlisle’s original explanation: DAWKINS, R., CARLISLE, T. Parental investment, mate desertion and a fallacy. Nature 262, 131–133 (1976). https://doi.org/10.1038/262131a0

This sentence should therefore be removed. I actually think your next few sentences (discussing the idea that many birds don’t coordinate incubation but do coordinate chick rearing) illustrates the idea that coordination might be more pronounced during chick rearing just fine. You could also discuss the increased energetic costs of chick rearing vs incubation and the increased constraints associated with dividing self-care and chick care, for which the literature is full of examples.

L122-123: It’s not enough to just discuss ‘patterns’ – please explain what you mean and give examples (e.g. lots of seabirds where foraging trips are longer during incubation than chick-rearing). As pointed out in the rebuttal, PLOS is a broad journal, so it is important to explain what this means behaviourally.

L125: ‘Runway’ is not a term I have encountered in the literature, and I doubt I’m alone – I suggest explaining as it was in the rebuttal letter i.e. parents may need time to synchronise to each other’s behaviour prior to breeding.

L126: Given there is no evidence (based on the rebuttal letter) I suggest removing; ‘familiarity’ is poorly defined in the literature anyway.

L157: It is essential to state that eggs can withstand neglect, and how long for. If eggs could only survive 10 minutes of neglect, then my original point that successful breeders are constrained to apparently coordinate would still stand. This is crucial to the rest of the analyses so needs to be stated explicitly and clearly.

L166: I still don’t understand what is meant by ‘habituation to the parental mode’ even with the rebuttal letter. This needs explanation – perhaps ‘increased within-pair synchronicity’?

L244: Please include explanation from rebuttal letter for why observer effects are not a concern. In the rebuttal it states ‘we made sure new observer would not differ’ – how, if there were no stats? Worth outlining in supplementary.

L254: Please include explanation from the rebuttal letter.

L271: See my main comments above about whether colony behaviour is negligible or not.

L272: I don’t think ‘and their constraints’ adds anything here, and it’s confusing. I suggest removing it.

L296: Please include explanation from the rebuttal letter.

L323-324: ‘Phase’ is used to mean something different for incubation and chick rearing. This is really confusing as they are

different metrics (continuous numerical vs categorical) and is misleading. Please rephrase.

L328: What is the distribution of the response variable, please state.

L360: State that the two chick rearing phases are ‘early’ and ‘mid’ (perhaps in brackets). There is a lot of reliance on readers remembering quite specific pieces of information, please restate these sorts of things when relevant to make it easier.

L364-368: This is still a very difficult sentence to follow and very long. Please rephrase.

L376-377: My confusion here arose because the phrasing suggests that there was a coordination index for both incubation

and chick rearing (‘In contrast to the incubation period, we decided to use for the chick rearing period the coordination index’), which I understand from the rebuttal letter is not correct. Please rephrase.

L384: Please include explanation from the rebuttal letter.

L506: I suggest some caveat e.g. behaviour is flexible between breeding stages, because you don’t show intra-individual flexibility and therefore can’t technically refute the ‘sealed bid’ model (e.g. parents could submit a fixed sealed bid for incubation and chick rearing separately, which doesn’t change – this is unlikely but we can’t say with certainty).

L544-545: In my original review, I did not mean that hormones don't play a role. My point is that in the discussion, you seem to be asking evolutionary/ultimate questions – i.e. what has driven the evolution of coordinated incubation? For example, you point out that coordination may evolve because it favours embryo development or because it helps to protect chicks prior to thermal independence.

Yes, hormones are likely to be a mechanism by which coordination is achieved – but *why* has this mechanism evolved? It is suggested that hormones explain why birds become more devoted to incubation over time – mechanistically this might be true, but it doesn't answer the 'why' question. Many birds fly south for winter - this could be driven by some hormonally mediated instinct, but that doesn't tell us why they do it in the first place.

It's fine to say that hormonal changes regulate behaviour, but I would put this somewhere else in the discussion and make clear that this is a mechanistic question. In fact, a better option might be to put this in the introduction to help show that levels of coordination change over time. E.g. something along the lines of: prolactin increases over the course of incubation, suggesting changes in parental care behaviour over time - but whether this increases parents’ propensity to incubate, and therefore their level of coordination, is unknown.

L565-567: Please include explanation from the rebuttal letter.

L572-573: It seems surprising to expect coordination to ‘reset’ after chick-rearing as outlined in the rebuttal; this needs further explanation (in the manuscript).

L582: I think perhaps ‘differences’ would make more sense– instability is still not clear.

L594: I suggest ‘reflect’ as ‘express’ suggests an active choice.

L595-596: Please include explanation from the rebuttal letter.

Reviewer #3: (No Response)

7. PLOS authors have the option to publish the peer review history of their article (what does this mean?). If published, this will include your full peer review and any attached files.

Reviewer #1: **Yes: **María Félix-Lizárraga

Reviewer #2: No

Reviewer #3: No

---

## [Author Response · Author response to Decision Letter 1]

7 Jun 2024

See attached file "Response to Reviewers"

---

## [Decision Letter · Decision Letter 2]

24 Jun 2024

Coordination of parental performance is breeding phase-dependent in the Dovekie (Alle alle), a pelagic Arctic seabird

PONE-D-23-40200R2

Dear Dr. Grissot,

We’re pleased to inform you that your manuscript has been judged scientifically suitable for publication and will be formally accepted for publication once it meets all outstanding technical requirements.

Kind regards,

Cord M. Brundage, D.V.M., Ph.D.

Academic Editor

PLOS ONE

Reviewers' comments:

Reviewer's Responses to Questions

**Comments to the Author**

1. If the authors have adequately addressed your comments raised in a previous round of review and you feel that this manuscript is now acceptable for publication, you may indicate that here to bypass the “Comments to the Author” section, enter your conflict of interest statement in the “Confidential to Editor” section, and submit your "Accept" recommendation.

Reviewer #1: All comments have been addressed

Reviewer #2: (No Response)

2. Is the manuscript technically sound, and do the data support the conclusions?

Reviewer #1: Yes

Reviewer #2: Yes

3. Has the statistical analysis been performed appropriately and rigorously? 

Reviewer #1: Yes

Reviewer #2: Yes

4. Have the authors made all data underlying the findings in their manuscript fully available?

Reviewer #1: Yes

Reviewer #2: No

5. Is the manuscript presented in an intelligible fashion and written in standard English?

Reviewer #1: Yes

Reviewer #2: Yes

6. Review Comments to the Author

Reviewer #1: Comments were addressed appropriately by the authors. The manuscripts its completed and ready . Thank you to the authors of the effort.

Reviewer #2: Thank you to the authors for their efforts in revising this manuscript; it is much improved as a result. I have just two very short comments. Firstly, in the abstract, L32 - the sentence claims that coordination was found to be 'condition-dependent'. This was not explored in the study, so please remove this. Secondly, throughout the manuscript, there are a few instances where the word 'the' appears before 'incubation' or 'chick-rearing' e.g. L116-117: 'During the incubation, parental coordination...' should be 'During incubation, parental coordination...'. Most of these are addressed but quite a few still remain that should be removed.

7. PLOS authors have the option to publish the peer review history of their article (what does this mean?). If published, this will include your full peer review and any attached files.

Reviewer #1: **Yes: **María Félix-Lizárraga

Reviewer #2: No

---

## [Editor Report · Acceptance letter]

18 Jul 2024

PONE-D-23-40200R2 

PLOS ONE

Dear Dr. Grissot, 

I'm pleased to inform you that your manuscript has been deemed suitable for publication in PLOS ONE. Congratulations! Your manuscript is now being handed over to our production team.

Kind regards, 

on behalf of

Dr. Cord M. Brundage 

Academic Editor

PLOS ONE